# HyperTRIBE uncovers increased MUSASHI-2 RNA binding activity and differential regulation in leukemic stem cells

Diu T. T. Nguyen[1], Yuheng Lu[2,3,7], Eren L. Chu[1,4,7], Xuejing Yang[1], Sun-Mi Park[1], Zi-Ning Choo[4], Christopher R. Chin [4], Camila Prieto[1], Alexandra Schurer[1], Ersilia Barin[1], Angela M. Savino[1], Saroj Gourkanti[1], Payal Patel[4], Ly P. Vu[5,6], Christina S. Leslie [2] & Michael G. Kharas [1✉]

The cell-context dependency for RNA binding proteins (RBPs) mediated control of stem cell fate remains to be defined. Here we adapt the HyperTRIBE method using an RBP fused to a *Drosophila* RNA editing enzyme (ADAR) to globally map the mRNA targets of the RBP MSI2 in mammalian adult normal and malignant stem cells. We reveal a unique MUSASHI-2 (MSI2) mRNA binding network in hematopoietic stem cells that changes during transition to multipotent progenitors. Additionally, we discover a significant increase in RNA binding activity of MSI2 in leukemic stem cells compared with normal hematopoietic stem and progenitor cells, resulting in selective regulation of MSI2's oncogenic targets. This provides a basis for MSI2 increased dependency in leukemia cells compared to normal cells. Moreover, our study provides a way to measure RBP function in rare cells and suggests that RBPs can achieve differential binding activity during cell state transition independent of gene expression.

[1] Molecular Pharmacology Program, Center for Cell Engineering, Center for Stem Cell Biology, Center for Experimental Therapeutics, Center for Hematologic Malignancies, Memorial Sloan Kettering Cancer Center, New York, NY 10065, USA. [2] Computational Biology Program, Memorial Sloan Kettering Cancer Center, New York, NY, USA. [3] Blavatnik Institute of System Biology, Harvard Medical School, Boston, MA 02115, USA. [4] Weill Cornell School of Medical Sciences, New York, NY 10065, USA. [5] Terry Fox Laboratory, British Columbia Cancer Research Centre, Vancouver, BC V5Z 1L3, Canada. [6] Molecular Biology and Biochemistry, Simon Fraser University, Vancouver, BC V5A 1S6, Canada. [7] These authors contributed equally: Yuheng Lu, Eren L. Chu. ✉email: kharasm@mskcc.org

While extensive research has revealed the crucial importance of transcriptional regulation, the role for post-transcriptional processes in the function of normal and cancer stem cells remains poorly defined. RNA binding proteins (RBPs) provide control of mRNA metabolism and translation of key regulators that mediate stem cells' self-renewal and cell fate decisions[1,2]. Moreover, mutations and aberrant expression of RBPs have recently been implicated in multiple types of cancer, demonstrating the crucial role for RBPs in tumorigenesis[3–9]. However, whether RBPs may have cell-type specific activity between different cellular states of normal stem cell differentiation or between normal and transformed contexts is not known. Understanding cell-specific targets provides a strategy for identifying unique cancer stem cell dependencies compared with normal cells, which is the key to developing new therapies.

Studying the molecular function of RBPs, as well as their cell-context dependency, requires the identification of their direct RNA targets in each cell type and in specific conditions. Standard approaches have relied heavily on native or cross-linking immunoprecipitation of RBPs followed by RNA-sequencing. They have been successfully employed to study RBP targets in embryonic stem cells, neural stem cells, and iPSCs, which can be obtained in a large number[10–14]. However, these techniques remain technically challenging for rare cells with limited input material such as adult stem cells. Here, we address a critical gap in our understanding of RBP targeting in stem cells. We adapted a recently developed method, HyperTRIBE[15–17] to identify direct RBP targets in normal hematopoietic stem cells (HSCs) and leukemia stem cells (LSCs).

In HyperTRIBE, the catalytic domain of the *Drosophila* ADAR (Adenosine Deaminase Acting on RNA enzyme) is fused with an RBP. This fusion protein leaves a "fingerprint" on the RBP RNA targets by marking the binding sites with a nearby A-to-G editing event. HyperTRIBE was originally developed in *Drosophila*[15,16] and was not yet proven to work in mammalian systems. We selected MSI2, an RBP previously found to be essential for maintaining self-renewal in LSCs and to contribute to normal HSC engraftment and cell fate decisions[18–20], to demonstrate the feasibility and application of HyperTRIBE in mammalian stem cells.

In previous studies, MSI2 targets were identified in two independent AML cell lines (NB4 and K562) using CLIP methods[19,21]. Although these strategies characterized a handful of validated direct MSI2 mRNA targets, they did not provide a comprehensive map of endogenous targets in stem cells nor address cell-type specific binding activity of MSI2. Furthermore, while *Msi2* knockout mice exhibit a modest reduction in blood cells and about 50% reduction in hematopoietic stem and progenitor cells (HSPCs), depletion of MSI2 severely reduced the frequency and activity of LSCs in both mouse and human systems. This indicates a significantly higher dependency and requirement for MSI2 in LSCs and development of leukemia[20,22–26]. The cause for this differential requirement for MSI2 function in LSCs and HSCs is not known.

In this study, we employ our adapted HyperTRIBE approach to investigate the cell-type specific requirement of the RBP MSI2 in LSCs and normal HSPCs. We first demonstrate that HyperTRIBE method efficiently identifies MSI2 mRNA targets in mammalian cells. We then globally map MSI2 mRNA binding network in HSCs and reveal MSI2 targeting program changes during differentiation into multipotent progenitors (MPPs). Furthermore, we find that RNA binding activity of MSI2 significantly increases in LSCs compared with normal HSPCs, which results in selective regulation of MSI2's oncogenic targets. Overall, this work suggests that RBPs can achieve cell-context dependent binding

activity, and demonstrates a strategy to study RBP functions in rare cells.

## Results

**MSI2-HyperTRIBE identifies MSI2 RNA targets in human cells.** HyperTRIBE was originally developed to map RBP targets in *Drosophila* cells[15–17]. In order to measure RBP targets in mammalian cells, we fused the human MSI2 with the catalytic domain of *Drosophila* ADAR (MSI2-ADA) carrying the hyperactive mutant E488Q previously described to increase editing[27]. Codon optimization was performed to maximize the expression of the fusion protein in human cells. To control for the background editing, we introduced an E367A catalytic dead mutation[28,29] in the ADAR domain (MSI2-DCD, Fig. 1a, Supplementary Fig. 1a). Overexpression of MSI2-ADA in the human AML cell line MOLM-13 resulted in a significant increase (over sixfold) in the number of A->G editing events and edit frequency on RNAs compared with the empty vector control (MIG) (Fig. 1b, c). Overexpressing the catalytic dead fusion MSI2-DCD did not lead to any increase in edit sites or frequency (Supplementary Fig. 1a, Fig. 1b, c), indicating that MSI2-ADA's increase in editing events is specifically due to its deaminase activity. These data suggest that we successfully adapted *Drosophila* Hyper-TRIBE to mammalian RBPs. Importantly, to take into account the background editing by these controls, when calculating the actual edit frequency at each site (now referred to as differential edit frequency or diff.frequency) we subtracted the mean edit frequency of MSI2-DCD and MIG from the mean edit frequency of MSI2-ADA.

We next assessed the reproducibility and the effect of overexpressing the MSI2-HyperTRIBE fusions on global gene expression (GE). Pair-wise correlation analysis of three independent experiments suggests that the edit frequency is highly reproducible (Pearson correlation coefficient $r > 0.8$, Supplementary Fig. 1b–d).

In contrast to CLIP based strategies, we found that the edit frequency is largely independent of the expression level of the target mRNAs (Supplementary Fig. 1e). Moreover, MSI2 and the fusion overexpression for 48 h did not lead to any major changes in the transcriptome of the cells suggesting that forced expression did not alter mRNA target abundance (Supplementary Fig. 1f–h). Overall these data indicate that the editing activity reflects MSI2 binding and that it can be used to reliably assess RBP binding.

To assess the accuracy of RNA target identification by the mammalian HyperTRIBE, we first mapped the binding sites to specific genes and compared with CLIP strategies. MSI2-HyperTRIBE identified 2056 target genes marked by 5244 significant edit sites in the human AML cell line MOLM-13. The majority of sites (~94%) were located in the 3′UTR region (Fig. 1d, Supplementary Data 1), which is consistent with previous studies[21,30]. To determine if MSI2-HyperTRIBE identifies a preferred binding sequence, we performed a de novo motif search using 200 bp sequences centered at the edit sites. We identified the known MSI2 binding motif (Fig. 1e) and confirmed that it was enriched within 250 bp of edit sites (Fig. 1f, Supplementary Data 2)[31,32]. In addition, the editing occurred either on or near sites that were directly bound by MSI2 as previously identified by CLIP (Fig. 1f)[21]. The top 255 genes with the highest differential frequency of at least 0.4 are positively correlated with genes upregulated upon MSI2 depletion in four human AML cell lines[18] (Fig. 1g). These targets also correspond to the top hits with highest number of peaks in our previous MSI2 HITS-CLIP analysis in the K562 cell line[19], (Supplementary Fig. 1i). Our results demonstrate that MSI2-HyperTRIBE efficiently identified direct MSI2 binding targets in mammalian cells.

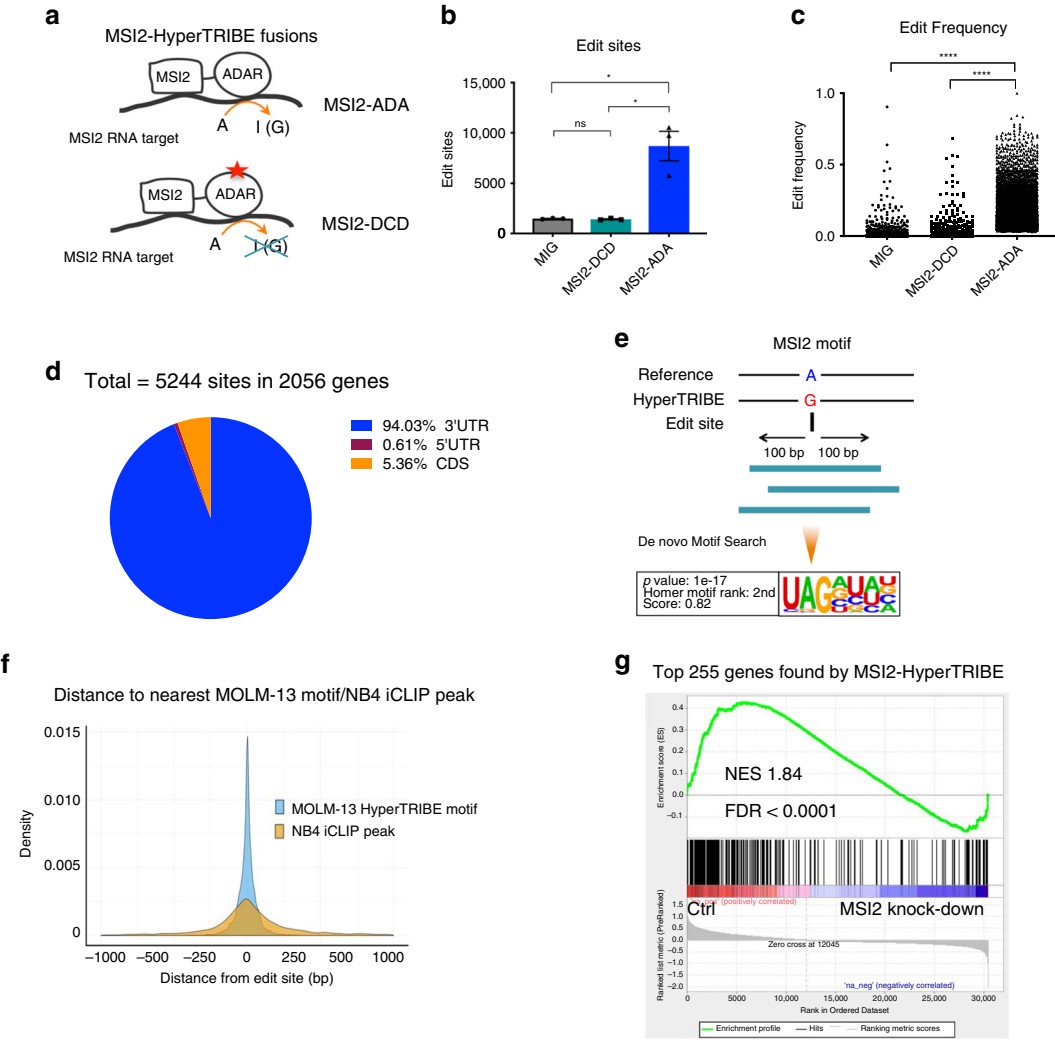

**Fig. 1 MSI2-HyperTRIBE identifies MSI2's direct mRNA targets in a human leukemia cell line. a** Schematic illustration showing the MSI2 protein fusion with the catalytic domain of hyperactive ADAR (MSI2-ADA) and the control fusion of MSI2 with the ADAR dead catalytic domain (MSI2-DCD). **b** Number of edit sites on mRNAs in MOLM-13 cells overexpressing MSI2-ADA or controls MSI2-DCD and empty vector (MIG). Data as means ± SEM of all the data points in three independent experiments. Two-tailed unpaired Student $t$ test; *$p < 0.05$. **c** Edit frequency on mRNAs in MOLM-13 cells overexpressing MSI2-ADA or controls MSI2-DCD and empty vector MIG. Only significant edit frequency (adjusted $p < 0.05$) are plotted. Data as means ± SEM of all the data points in three independent experiments. Unpaired Mann–Whitney test; ****$p < 0.0001$. **d** Total number of MSI2-HyperTRIBE significant edit sites, target genes, and distribution of sites on the genes in MOLM-13 cells from three HyperTRIBE experiments. **e** Illustration of selected window size surrounding edit sites for de novo motif analysis and the results showing enrichment of a consensus sequence that matches previously identified MSI2 motif. **f** Probability density function (pdf) plot showing the spatial distribution of distance from edit sites to the nearest MSI2 motifs found in **d** (light blue) and from edit sites to nearest NB4 iCLIP peak (dark yellow). **g** GSEA analysis shows that top targets found by MSI2-HyperTRIBE (255 genes with diff. frequency ≥ 0.4) are enriched among genes that are differentially expressed in MSI2-depleted human AML cell lines compared with controls (data in Kharas et al.[18]). $y$-axis shows enrichment score of the 255 geneset. The black bars on the $x$-axis show the genes in the MSI2-depleted RNA-seq ranked list, with log2fc(control/knockdown) value high to low running from left to right. *NES* normalized enrichment score.

Since multiple sites were found on the same RNA target, we looked to see if there was a pattern of clustered binding. To decide on a suitable window size for clustering edit sites, we compared the enrichment of MSI2 motifs in windows of fixed size around significantly edited sites (true sites) with windows of the same size around non-significantly edited sites (background). Using a Fisher's test, we determined that ±17 bp is the largest window such that the motif enrichment was significantly greater around true sites compared with background. We therefore clustered nearby edit sites falling within this window size and found that the majority of clusters (87%) contain only single sites, suggesting that MSI2 binds RNA and then ADAR edits mainly at these discrete sites (Supplementary Fig. 2a, b). Therefore, the majority of MSI2-HyperTRIBE's edit sites represent MSI2 binding.

To further rule out the potential of non-specific binding by MSI2-HyperTRIBE, we performed additional controls using a fusion of ADAR with MSI2 lacking RNA binding activity, as well as HyperTRIBE with ADAR domain alone without MSI2. To this end, we overexpressed the catalytic domain ADAR alone (ADA only) and ADAR fused with MSI2 lacking both RRMs (RNA Recognition Motifs), RRM(del)MSI2-ADA, or with MSI2 mutated at five amino acids in both RRM domains that are crucial for RNA binding activity, RRM(mut)MSI2-ADA (Supplementary Fig. 3a)[33]. Our analysis found that ADAR alone and the mutant fusions have low editing frequency and produce only a few significant edit sites (52 sites for ADA only, 18 for RRM(del)MSI2-ADA and 20 for RRM(mut)MSI2-ADA) compared with MSI2-ADA fusion (5244 significant sites) (Supplementary

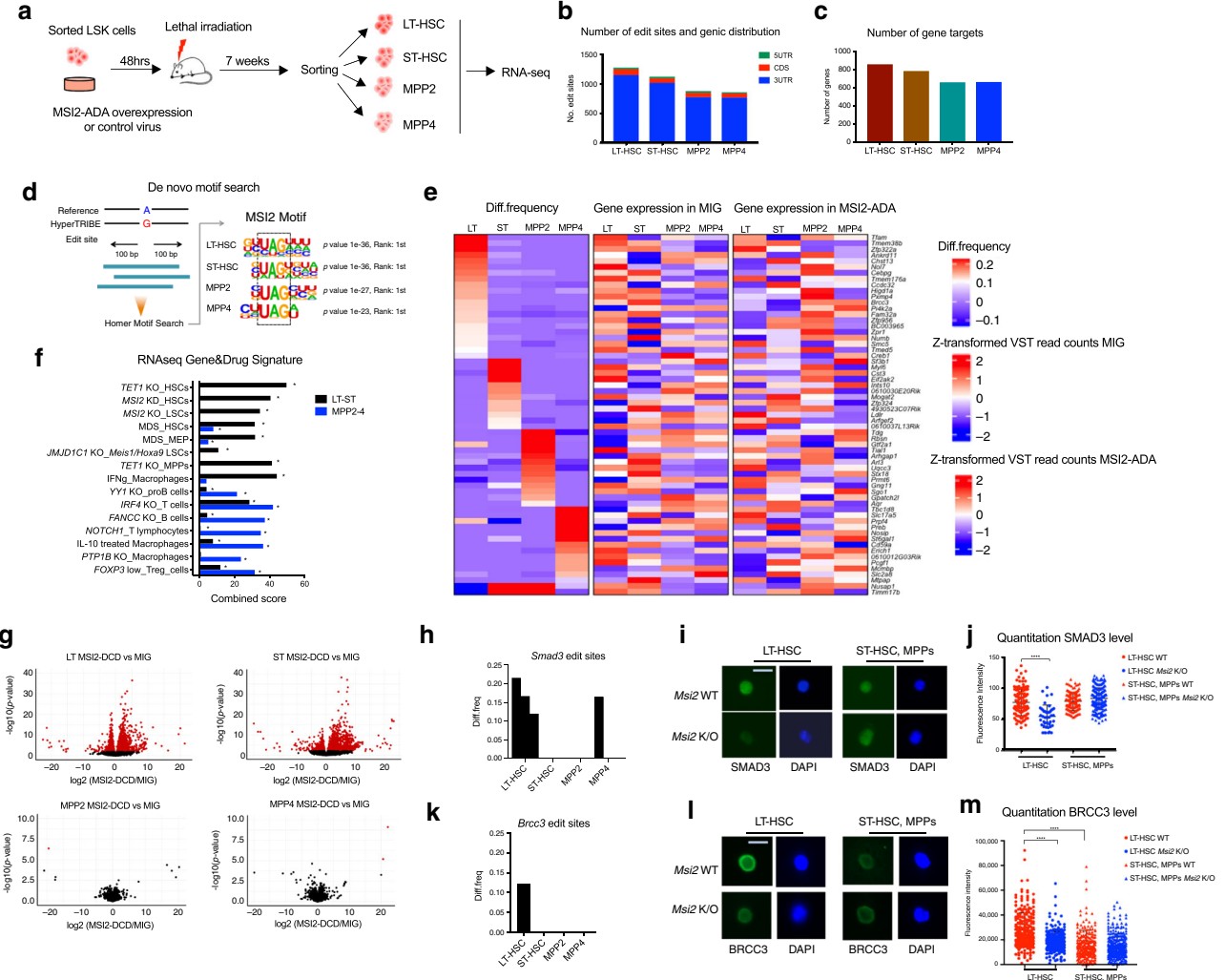

**Fig. 2 Cell context MSI2 binding during hematopoietic stem cell differentiation. a** Schematic illustration of MSI2-HyperTRIBE in HSPCs in vivo. $n = 2$ independent experiments. **b** Number of MSI2-HyperTRIBE significant edit sites and their genic distribution in four compartments of HSPCs. **c** Number of target genes with sites (described in **b**) in HSPCs. **d** De novo motif search showing enrichment of MSI2 motif in all four populations of HSPCs. **e** Clustering of diff.frequency for target genes across cell types (left panel). Only genes more significantly edited (beta-binomial test) in one cell type versus all others are plotted. Relative gene expression of each target, in same row order as diff.frequency heatmap, in control cells MIG (middle panel) and for MSI2-ADA overexpressing cells (right panel). LT:LT-HSC; ST:ST-HSC. **f** RNA-seq Gene and Drug Signature analysis for MSI2 targets in LT and ST HSCs (LT-unique, ST-unique and Shared LT-ST) compared with targets in MPPs (MPP2-unique, MPP4 unique and Shared MPP2-MPP4). Asterisks indicate FDR < 0.05. **g** Differential expression (DEseq2) analysis of MSI2 overexpression in four HSPCs populations. Red dots represent genes with significant differential expression in MSI2-DCD versus MIG control. **h** Editing occurs on *Smad3* mRNAs at three sites in LT-HSC, 0 sites in ST-HSC and MPP2 and one site in MPP4. Each bar represents one site. **i** Representative images of immunofluorescence analysis (IF) showing SMAD3 signal in LT versus ST, MPPs. Scale bar 5 μm. **j** Quantitation of SMAD3 IF signal from **i**. $n = 125$; 45; 130, and 203 cells for LT *Msi2* WT; KO; ST, MPPs *Msi2* WT and KO, respectively. Data as mean ± SEM. Unpaired Student $t$ test, ****$p < 0.0001$. **k** Editing occurs on *Brcc3* mRNAs only in LT-HSC and not in other populations. Each bar represents one site. **l** Representative IF images showing BRCC3 signal in LT versus ST and MPPs. Scale bar 5 μm. **m** Quantitation of BRCC3 IF signal from **l** in *Msi2* WT and *Msi2* K/O. $n = 258$; 263; 216 and 295 cells for LT *Msi2* WT; KO; ST, MPPs *Msi2* WT and KO, respectively. Data as mean ± SEM. Unpaired Student $t$ test, ****$p < 0.0001$.

Fig. 3b–d). These data indicate that MSI2 and its RRMs provide the cellular binding specificity for ADAR editing.

**Cell-context dependent RNA binding activity of MSI2 in HSPCs.** Given that MSI2 is highly expressed in both HSCs and MPPs and that loss of MSI2 results in a loss of quiescence and reduced self-renewal[18,19,21], we hypothesized that there could be differential targets in HSCs compared with MPPs. Thus, we tested if HyperTRIBE can be applied to HSCs and MPPs by transducing MSI2-ADA, MSI2-DCD, or empty vector controls into Lin-, Sca1+, c-Kit+ cells (LSKs) isolated from C57/BJ6 mice. We then transplanted these cells into lethally irradiated mice and after they were engrafted, long-term HSCs (LT-HSCs), short-term HSCs (ST-HSCs), multipotent progenitors MPP2 and MPP4 were isolated, followed by RNA-seq (Fig. 2a, Supplementary Fig. 4a). We were able to detect 1273 edit sites in LT-HSCs, 1126 sites in ST-HSCs, 879 and 862 sites in MPP2s and MPP4s, respectively (Fig. 2b). These edit sites represented 856 gene targets in LT-HSCs, 782 genes in ST-HSCs, 658 genes in MPP2, and 661 in MPP4 (Fig. 2c, Supplementary Data 1). Furthermore, despite equivalent expression of the MSI2-HyperTRIBE fusions, we observed more edit sites (~1.4–1.5 fold), gene targets (~1.2–1.3 fold), and more targets marked with at least two sites in HSCs

compared with MPPs (Fig. 2b, c, Supplementary Fig. 4b–d). These data suggest that MSI2 binding activity is modestly increased in HSCs compared with MPPs.

To determine if MSI's binding sites were conserved in HSPCs and if they changed during differentiation, we performed de novo motif analysis. Similar to the MOLM-13 cells, the same MSI2 motif was found to be the most enriched in all populations (Fig. 2d, Supplementary Data 2). These data confirm that the edit sites marked MSI2 binding sites and demonstrate that Hyper-TRIBE can identify an RBP's RNA targets in limited cell numbers.

We then investigated if and how the MSI2 binding changed when HSCs differentiated into more committed progenitors. Clustering of gene targets by differential edit frequency (diff. frequency) across cell types revealed a group of mRNA targets bound by MSI2 in all four states of HSPCs with no significant difference in diff.frequency (vs controls) between populations (beta-binomial test, FDR ≥ 0.1) (Supplementary Fig. 4e). In addition, there are subsets of transcripts that are bound only in a specific state (unique groups, Fig. 2e) with diff.frequency (vs controls) significantly different in one state compared with all other states (beta-binomial test, FDR < 0.1; $p$ value < 0.05). Importantly, we did not observe a similar pattern of mRNA expression of the targets (middle and right panel, Fig. 2e), suggesting that the majority of differential binding activity at different states of HSPCs is not simply a consequence of the differential abundance of mRNA transcripts. These data support the concept that RBP activity and target engagement depends on cell states.

We then hypothesized that the abundance and target spectrum could also result in altered biological functions of the shared and specific targets in HSCs versus those in MPPs. Thus, we performed gene pathway enrichment analysis using the ENRICHR program[34] for targets specific and shared in LT and ST-HSC versus targets in MPPs (489 vs 298, Supplementary Figs. 4f, 5a, Supplementary Data 3). We found that HSC targets are highly enriched for stem cell programs, such as HSCs, MDS and LSCs; whereas MPP targets are enriched for lineage-specific programs, such as macrophages, T cells and B cells (Fig. 2f, Supplementary Fig. 5b, Supplementary Data 3). In addition, gene ontology (GO molecular functions) analysis indicates that HSC targets enriches for RNA binding, kinase binding and ubiquitin ligase activity whereas MPP targets are involved in RNA polII coactivator binding (Supplementary Fig. 5c, d, Supplementary Data 4). These data indicate that MSI2 switches its binding targets away from HSC-related pathways toward differentiation-associated pathways as the cells differentiate to MPPs.

Previous studies, using normal and MDS mouse models, found that inducible overexpression of MSI2 results in the expansion of HSPC populations[18,21,23,24,35], but the overexpression impact on specific subsets within the HSPC compartments remains unclear. Thus, we compared the GE profile of MSI2 overexpression (MSI2-DCD) to control (MIG) in HSCs and in MPPs. MSI2 overexpression resulted in significant changes in the transcriptome in LT and ST HSCs but not in MPPs, suggesting that MSI2 impacts HSCs differentially compared with MPPs (Fig. 2g). Notably, most of these genes with expression changes were not direct MSI2 targets (~6% 195 out of 2972 differentially expressed genes in LT; 113 out of 2047 in ST HSCs) (Supplementary Fig. 5e). These results suggest that although HSCs have a modest increase in MSI2 binding compared with MPPs, it results in a large transcriptional effect. However, this effect is indirect and likely through its small subset of direct binding targets in HSCs.

Our previous study found that MSI2 directly controls TGFB signaling output[19]. Based on our MSI2 differential binding activity, we examined *Smad3*, a direct target in the TGFB signaling pathway that was found by HITS-CLIP in K562 cells and has reduced protein abundance in HSCs upon *Msi2* depletion[19]. HyperTRIBE identified that MSI2 bound more efficiently to *Smad3* transcripts in LT-HSCs than in ST-HSCs, MPP2, and MPP4 (Fig. 2h). This corresponded to a decrease in total SMAD3 and phosphorylated SMAD3 protein in LT-HSCs but not in ST-HSCs and MPPs upon *Msi2* knockout (Fig. 2i, j and Supplementary Fig. 5f, g). In addition, among 21 targets that are more significantly edited (shown in the heatmap, Fig. 2e) in LT-HSCs versus all other populations, *Brcc3* or BRCA1/BRCA2 containing complex 3, has been reported to be mutated in myelodysplasia syndrome (MDS) and in de novo AML[36,37]. These mutations are associated with clonal hematopoiesis, which suggests that *Brcc3* plays a key functional role in HSCs. *Brcc3* is uniquely targeted by MSI2 in LT-HSCs but not in more committed progenitors (Fig. 2k). We therefore chose this candidate for validation as a novel HSC target. Similar to SMAD3, MSI2 depletion led to significant reduction of BRCC3 abundance in LT-HSCs but not in ST-HSCs, MPP2s and MPP4s (Fig. 2l, m). Of note, the mRNA level of *Smad3*[19] or *Brcc3* (Supplementary Fig. 5h) was unaffected by MSI2 depletion suggesting that SMAD3 and BRCC3 translation was being controlled specifically in LT-HSCs compared with ST-HSCs and MPPs. Moreover, LT-HSC have increased BRCC3 protein abundance without a significant difference in expression *Brcc3* transcript compared with ST-HSCs and MPPs (Fig. 2m and Supplementary Fig. 5i). The equivalent transcript abundance of *Smad3* was also observed between these two populations (Supplementary Fig. 5i). Overall, our data indicate that despite similar abundance of MSI2 and its RNA targets, MSI2 can differentially control its targets' protein abundance during hematopoietic differentiation.

**Increased MSI2 RNA binding activity in LSCs versus HSPCs.** Although MSI2 has been demonstrated to play an important role in both HSPCs and LSCs, it remains unclear why LSCs are more dependent on MSI2 compared with normal cells. Thus, we expressed the MSI2-ADA fusion and controls in LSCs (c-Kit[hi] cells) isolated from quaternary MLL-AF9-dsRed mice and normal HSPCs (LSKs). Our analysis detected over 12,000 sites located in 2865 genes in LSKs. Strikingly, we observed 2.5 times more edit sites (30,701 vs 12,071 sites) and 1.4 times more target genes (4162 vs 2865 genes) in LSCs despite a lower expression of MSI2-ADA fusion and endogenous MSI2 in LSCs compared with LSKs (Fig. 3a, Supplementary Fig. 6a, b). In addition, over 60% of MSI2 targets identified by HyperTRIBE in human leukemia cells are conserved in murine leukemia (Supplementary Fig. 6c, Supplementary Data 1). These data suggest that MSI2 has increased target engagement in leukemia versus normal cells.

To assess the differences in MSI2 binding in LSCs versus normal cells, we examined the location of editing, the shared and cell-specific sites. Consistent with our previous results, almost all the edit sites (~93%) were located in 3′UTR and the MSI2 binding motif was the most enriched consensus sequence around the edit sites in both LSKs and LSCs (Fig. 3a, Supplementary Fig. 6d–f, Supplementary Data 1 and 2). The vast majority of sites (nearly 80%) and genes (over 87%) marked by MSI2-ADA in LSKs were also found in LSCs, and the number of targets bound by MSI2 only in LSCs (1656 LSC unique targets) was approximately five times higher than those bound only in LSKs (359 LSK unique targets) (Fig. 3b, Supplementary Fig. 6g, Supplementary Data 1). Moreover, there are more edit sites per MSI2 target in LSCs compared with LSKs (Supplementary Fig. 6h, i) and at the shared sites, we found that they were edited at higher frequency in LSCs than in LSKs (Fig. 3c). These data suggest that despite similar

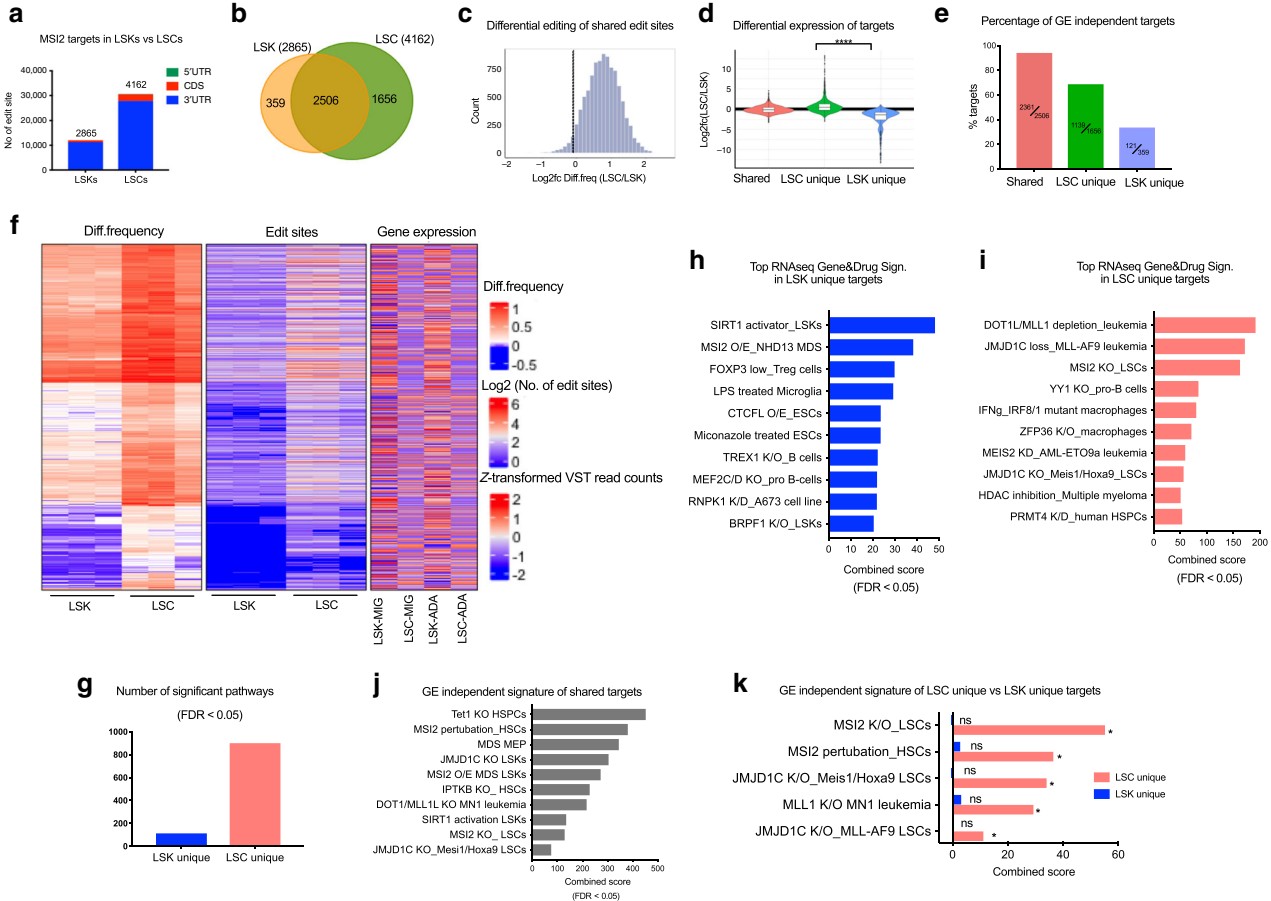

**Fig. 3 Increased MSI2 RNA binding activity in LSCs. a** Number of MSI2-HyperTRIBE significant edit sites and their distribution on genes in LSKs and LSCs. Number of target genes in each cell type is shown on top of the bars. $n = 3$. **b** Overlapping of target genes in LSKs and LSCs: 2506 shared, 1656 LSC unique targets, and 359 LSK unique targets. **c** Differential editing of shared sites, represented by Log2 fold change of diff.frequency in LSCs and in LSKs. **d** Violin plot presenting log2 fold change of gene expression in LSCs and LSKs (overexpressing MIG) of shared targets, LSC unique targets ($n = 1651$) and LSK unique targets ($n = 359$). One-sided Wilcoxon test. ****$p < 0.0001$. Plot center lines show the median, box limits denote upper and lower quartiles, whiskers represent 1.5× interquartile range and individual points show outliers. **e** Percentage of gene expression (GE) independent targets in shared, LSC unique and LSK unique target groups from **b**. **f** Clustering of diff.frequency for top gene targets with diff.frequency of at least 0.6 in LSKs and LSCs (left panel). Only genes with diff.frequency significantly different (LSC vs LSK, beta-binomial test), are plotted. Matched number of edit sites for each target (per row) (the middle panel) and corresponding expression level in LSKs versus LSCs (right panel). **g** Total number of significant RNA-seq Gene and Drug signatures (FDR < 0.05) enriched in LSK and LSC unique targets. **h** Top significant RNA-seq Gene and Drug signatures enriched in LSK unique targets (359 genes) using ENRICHR analysis. FDR < 0.05 for all indicated pathways. **i** Top significant RNA-seq Gene and Drug signatures enriched in LSC unique targets (1656 genes) using ENRICHR analysis. FDR < 0.05 for all indicated pathways. **j** Gene expression (GE) independent RNA-seq Gene and Drug signatures of shared targets in LSKs and LSCs. Full list of shared target genes in **b** is filtered with log2fc (LSC-MIG/LSK-MIG) ≤ 1.2. **k** GE independent signature of RNA-seq Gene and Drug signatures of LSC unique and LSK unique targets. *FDR < 0.05.

expression between normal cells and leukemia cells the activity of MSI2 is increased in LSCs compared with normal cells.

To assess whether the elevated RNA binding activity of MSI2 in LSCs is due to higher abundancy of the targets, we carried out differential expression analysis comparing expression of mRNAs between LSCs and LSKs. We observed that almost all shared (~94%) and the majority (~69%) of LSC unique targets have comparable expression in both cell types or lower expression in LSKs (log2fc LSC/LSK ≤ 0.26 or FDR ≥ 0.05 no significant difference) whereas the majority (~66%) of LSK-specific targets were expressed more highly in LSKs (log2fc LSC/LSK ≤ −0.26) (Fig. 3d, e). Thus, RNA transcript abundance could explain a proportion but not the majority of the differential binding activity in LSCs.

To determine the significant differences in MSI2 binding in LSCs, we clustered the differential edit frequency of targets in both cell types. We observed the elevated editing in LSCs versus

LSKs even in the most highly edited targets (≥0.6 diff.frequency) as shown by an increase in both diff.frequency and number of edit sites (Fig. 3f). Importantly, for the majority of targets the mRNA expression could not simply explain this increased editing in leukemia compared with normal cells (right panel, Fig. 3f). Nevertheless, to further eliminate expression bias, we restricted the clustering to targets with comparable or lower expression in LSCs (vs LSKs) and still observed the same pattern of increased RNA binding in LSCs compared with LSKs (Supplementary Fig. 6j). Of note, the overexpression of MSI2-ADA and MSI2-DCD fusions for this short time course (48 h) did not result in significant changes in the transcriptome of both cell types (Supplementary Fig. 6k–p). These data suggest that MSI2 binding activity is elevated in LSCs versus LSKs through mechanisms independent of mRNA expression.

Next, we wanted to understand how differential RNA binding activity of MSI2 in LSCs compared with LSKs influences MSI2's

known functional pathways. Gene pathway analysis by ENRICHR revealed nearly 9 times more significant pathways enriched in the LSC unique targets versus the LSK unique targets (900 vs 113, FDR < 0.05) (Fig. 3g). Top LSK-specific signatures include normal embryonic stem cell related programs, hematopoietic stem cells and progenitors programs, while MSI2 controlled pathways and MLL-AF9 AML leukemia are amongst the most enriched signatures in LSC-specific targets (Fig. 3h, i, Supplementary Data 3). This is in accordance with our previous study, which demonstrates that MSI2 maintains the mixed-lineage leukemia (MLL) self-renewal program by controlling the translation of critical MLL regulated transcription factors such as *Hoxa9*, *Ikzf2* and *Myc* in myeloid leukemia[20]. In addition, gene ontology (GO Biological Processes) identified pathways related to RNA metabolism and protein transport and processing as well as translational regulation in LSC-specific targets while it did not find any significant biological processes in the LSK-specific targets (Supplementary Fig. 6q and Supplementary Data 4).

To investigate whether this is due to background cell-type specific expression of the targets, we performed gene enrichment analysis with only gene-expression (GE) independent targets (log2fc ≤ 0.26 or FDR ≥ 0.05 no significant difference, shown in Fig. 3e) for Shared, LSK unique and LSC unique groups. We found that the GE independent shared targets, the majority of which have higher binding to MSI2 in LSCs versus LSKs, are enriched for both normal HSPC-related as well as MLL-AF9 leukemia programs (Fig. 3j). Remarkably, MSI2 controlled pathways in LSCs and MLL1-HOXA9-MEIS1 leukemia programs were selectively enriched in GE independent LSC unique targets, which are expressed at the same or lower level in LSKs (Fig. 3k, Supplementary Data 3). Our results reveal that MSI2 not only enhances its RNA binding activity in LSCs versus LSKs overall, but also interacts more with genes regulated by the MLL leukemia programs in LSCs.

**Differential regulation of MSI2 targets in LSCs**. We then hypothesized that MSI2 differential binding to targets in the MLL program results in a specific effect on the abundance of the targets upon MSI2 perturbation in LSCs, compared with LSKs. To test our hypothesis, we looked at *Hoxa9*, *Ikzf2*, and *Myc*, our previously established MLL and MSI2 downstream targets as well as key transcription factors in hematopoiesis and leukemogenesis. We found that *Hoxa9* and *Ikzf2* 3′UTRs was substantially marked by MSI2-ADA (Fig. 4a, b). Although *Myc* was previously detected by CLIP and RIP approaches, we did not find any editing in *Myc* transcripts in all cell types in this study. This might be due to the rapid turnover of *Myc* mRNAs[9,38,39] and the stable interaction required for editing or because MSI2 does not actually bind *Myc* directly. However, we detected MSI2's interaction at *Myb*, a well-known upstream regulator of *Myc* and a key transcription factor in hematopoiesis as well as a driver of MLL related and non-related leukemia[40–45] (Fig. 3c).

We then confirmed the edit sites are indeed regulatory binding sites of MSI2 by a reporter assay with *Hoxa9* and *Myb*, which have relatively short 3′UTRs (Supplementary Fig. 7a, b). Interestingly, *Hoxa9*, *Ikzf2*, and *Myb* are less edited in LSKs as demonstrated by the fewer number of sites and lower differential edit frequency (Fig. 4a, c). Importantly, depletion of *Msi2* resulted in a significant reduction in protein, without changes in mRNA, of *Hoxa9*, *Ikzf2*, and *Myb*, in LSCs but not in LSKs (Fig. 4d, e, Supplementary Fig. 7c–e). Notably, HOXA9, IKZF2, and MYB abundance is modestly higher in LSCs compared with LSKs (Supplementary Fig. 7f). These data indicate that MSI2 is more required in LSCs to maintain the expression of these targets. Based on our results, we propose a model in which MSI2

increases interaction with its mRNA targets in LSCs, and therefore MSI2 ablation selectively affects the protein abundance of these targets in LSCs compared with normal LSKs. These data suggest that the increased RNA binding activity may explain the enhanced requirement of MSI2 in LSCs compared with LSKs.

**Discussion**
Although multiple studies have identified RBP mRNA targets in embryonic stem cells, pluripotent stem cells and neural stem cells isolated from embryos, which exist in large quantity[10–14,46], global mapping of RBP targets in rare cells such as adult normal and cancer stem cells has been hampered due to limited input material. The standard methods (RNA-IP and CLIPs including HITS-CLIP, iCLIP, eCLIP and sCLIP) require typically 5–20 millions of cells[47–50]. The irCLIP method for low input material requires 20,000–100,000 cells[51]. However, all of these CLIP methods require cross-linking and RBP immunoprecipitation (IP) which could result in either lost targets or the capture of nonspecific targets. In this study, we have successfully adapted the HyperTRIBE method, originally developed in *Drosophila*[15–17], for identification of RBP targets in mammalian cells. Utilizing our adapted HyperTRIBE method, we have obtained direct mRNA targets of an RBP in a human AML cell line and in mouse normal and transformed hematopoietic stem and progenitor cells. This method uses between 0.5 million cells (for MOLM13) to 360 cells (for LT-HSC) and does not need any cross-linking, IP, or labeling steps. We show in all of the cell types used in our study that this approach accurately captures the known binding motif of MSI2 in stem cells, an RBP that has been studied in various systems. Moreover, our data correlate well with previous studies that mapped MSI2 binding sites using immunoprecipitation techniques and we further validate the targets by genetic studies.

A-to-I editing by endogenous adenosine deaminase ADAR enzymes exists in cells to regulate RNA life cycle. This prompts the question whether the high expression of exogenous ADAR in the RBP-ADAR fusion artificially affects the expression and processing of target RNAs. We address this question by analyzing differential expression (DESeq2) for cells expressing MSI2-ADA compared with those with empty vector (MIG). Our analysis shows that there is little change in the transcriptome of MOLM13, LSKs, and LSCs expressing MSI2-ADA after 48 h of transduction. For in vivo HyperTRIBE in HSPCs, which took 7 weeks for transplantation and engraftment of cells expressing MSI2-ADA, we observed dramatic changes in transcriptome of LT-HSC and ST-HSC but not MPP2 and MPP4. Of the genes significantly changed upon MSI2-ADA expression, the majority is due to MSI2 overexpression, which is consistent with previous studies demonstrating a role of MSI2 in HSCs[18,21,23].

Although MSI2 binding sites have previously been identified in cell lines using alternative approaches, MSI2 binding in HSPCs and LSCs has never been characterized. Using HyperTRIBE, we are now able to assess the cell context specific MSI2 binding program for rare cell types including hematopoietic stem cells, MPPs, and leukemic stem cells. Importantly, our results demonstrate that RBP–RNA interactions are highly cell-context dependent even in closely related cell types. Although previous work has started addressing this question using in vitro differentiation culture[46,52], extensive and systematic studies are needed to assess RBP activity in rare cells during fate switches. Using our optimized HyperTRIBE method, we revealed that MSI2 has differential binding activity at different states of HSPCs and in LSCs in a target GE independent manner. Moreover, we found that the enhanced RNA binding activity of MSI2 leads to differential regulation, e.g., at *Hoxa9*, *Ikzf2*, and *Myb* targets, in LSCs versus LSKs, which provides a possible explanation for the differential

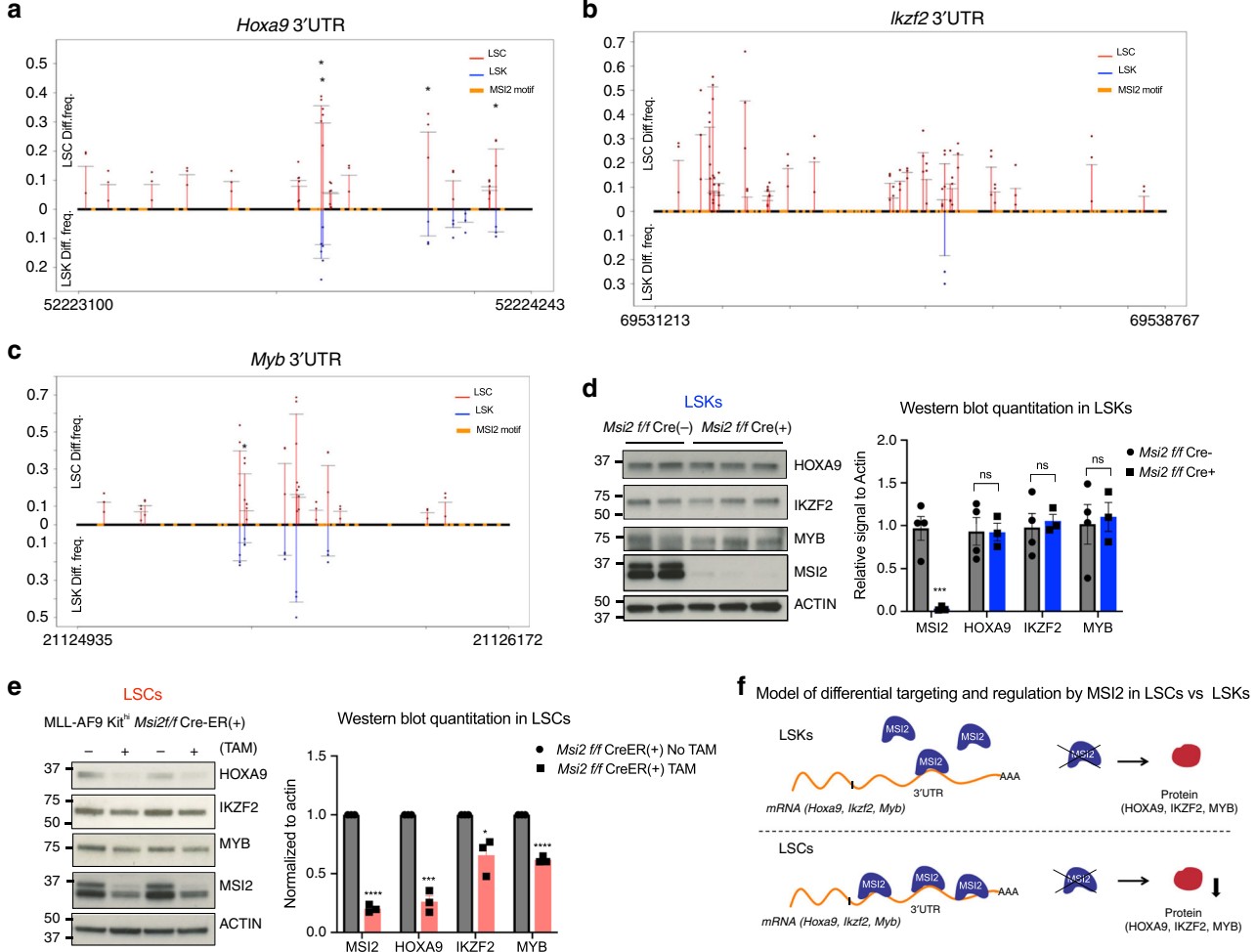

**Fig. 4 Differential control of MSI2 targets in LSCs compared with normal LSKs.** Diff.frequency at various sites identified by MSI2-HyperTRIBE in *Hoxa9* 3′ UTR (**a**), *Ikzf2* 3′UTR (**b**) and *Myb* 3′UTR (**c**) in LSKs and LSCs. Numbers on the *X*-axis is the start and end of 3′UTR. Data presented as the mean values from three independent HyperTRIBE experiments. Significant difference is determined by beta-binomial test. * adjusted $p < 0.1$ **d** Representative immunoblot images and quantitation showing no significant change in HOXA9, IKZF2 and MYB protein expression upon *Msi2* knockout in LSKs after 3 weeks of pIpC treatment in *Msi2* f/f Cre(−) and Cre(+) mice. Each data point is an independent treated mouse. Data are presented as mean ± SEM. Two-sided unpaired Student *t* test. ***$p < 0.005$. ($p = 0.002$ for MSI2). **e** Representative Immunoblot images and quantitation showing significant decrease in HOXA9, IKZF2 and MYB protein expression upon *Msi2* knockout at 68 h after TAM treatment in MLL-AF9 *Msi2* CreER(+) LSCs. Each data point is an independent treated mouse. Data are presented as mean ± SEM. $N = 6$ independent experiments for HOXA9 and IKZF2, $n = 3$ independent experiments for MYB. Two-sided paired Student *t* test. **$p < 0.01$, ***$p < 0.001$; ****$p < 0.0001$. ($p = 0.000002$ for MSI2, $p = 0.00015$ for HOXA9, $p = 0.019$ for IKZF2, $p = 0.000019$ for MYB). **f** Schematic depiction of MSI2 elevated RNA binding and reduction of target protein expression upon MSI2 ablation in LSCs, but not in LSKs.

requirement of MSI2 in leukemia compared with normal hematopoiesis.

Furthermore, it remains to be elucidated (1) how MSI2 achieves more binding to mRNA targets in LSCs even without upregulating MSI2 expression; and (2) why MSI2 controls protein abundance of its mRNA targets (e.g., *Hoxa9*, *Ikzf2,* and *Myb*) in LSCs but not in normal HSPCs. One possibility is that other RBPs that share a similar binding motif might compete for the same binding sites with MSI2 in LSKs. Alternatively, post-translational modifications on MSI2 or other RBPs could result in the increased binding. Moreover, multiple RBP-driven regulation pathways, including MSI2's, may coordinate to control translation process of their shared targets. Cancer cells often alter or lose multiple pathways and thus might become uniquely dependent on MSI2 regulation. Therefore, LSCs recruit more MSI2 to its targets rather than different RBPs as in normal LSKs. As a consequence, the regulation of the target expression is now more dependent on MSI2. Regardless of the exact mechanism, our data

support a leukemia-specific role for MSI2 and provide further rationale for targeting MSI2 in leukemia cells in patients that have equivalent expression of MSI2 as compared with normal cells. Our data provide a key resource for further studies on the mechanisms of RBP regulation in rare cells such as stem cell populations.

## Methods

**Animal research ethical regulation statement.** All animal studies were performed on animal protocols approved by the Institutional Animal Care and Use Committee (IACUC) at Memorial Sloan Kettering Cancer Center.

**Plasmid constructs.** MSI2-ADA fusion was constructed by fusing the human MSI2 CDS to the A-I deaminase domain of the *Drosophila* enzyme ADAR containing a hyperactive mutant E488Q[15], with a linker (the region from Y268 to the deaminase domain). The inactive ADAR catalytic mutant control MSI2-DCD was generated by mutating Glutamic acid E367 to Alanine in the deaminase domain[28,29], using site-directed mutagenesis (Agilent #200523). Both constructs were codon-optimized for expression in human cells before gene synthesis and

cloning into MSCV-IRES-GFP (MIG) vector. The sequence of these constructs are provided in the supplementary information (Supplementary Methods). After Sanger sequencing, we found that there was additional unexpected mutation, N495S, in the ADAR catalytic domain of the MSI2-DCD. However, this does not affect the fusion expression and we confirmed by the data in MOLM13 that the MSI2-DCD containing both E367 and N495S is catalytically inactive of A-to-I editing. RRM(del)MSI2-ADA was generated by removing both RRM1 and RRM2 of MSI2. To create RRM(mut)MSI2-ADA, we synthesized the fusion with RRM1 containing mutations F24A, R62A, F66A and F223A, F155A mutations on RRM2. To create ADA only construct, we removed MSI2 from the fusion MSI2-ADA. All of the contructs were fused with 2xFlag tags.

**Retroviral production and transductions**. Retroviral packaging of all expression constructs was performed in 293T cells as previously prescribed[53]. Retrovirus was kept at 4 °C and used within 2 weeks of production.

**MSI2-HyperTRIBE in MOLM-13 cell line**. MOLM-13 cells (obtained from ATCC) were cultured in RPMI 10% FBS 1%L-Glutamine PenStrep. Cells were infected with virus expressing MSI2-ADA, MSI2-DCD, or MIG controls at 1:1 ratio (v/v) cell: virus at 0.5 million cells per mL (final density). Spinoculation was done with 10 μg/mL polybrene (Millipore #TR-1003-G) at 768 g for 1 h at 32 °C. Cells were incubated for 48 h and then sorted by flow cytometry for GFP positive. At least 0.5 million GFP positive cells were used for RNA extraction and sequencing.

**MSI2-HyperTRIBE in HSPCs**. Bone marrow cells from 6 to 8-week-old C57BL/6 strain were processed for c-Kit enrichment by incubation with 50 μl of MACS CD117/c-Kit beads per mouse and then run on an AutoMACs (Miltenyi Biotec) following the manufacturer's instructions. Cells were stained with Lineage antibody cocktail including CD3 (Fisher #15-0031-83), B220 (ebioscience #15-0452-83), CD4 (Fisher #5013997), CD8 (ebioscience #15-0081-83), Gr-1 (ebioscience #15-5931-82), Ter119 (ebioscience #15-5921-83) (all conjugated with PE-Cy5), CD117-APC-Cy7 (Biolegend #105826), Sca-1-Pacific Blue (Biolegend #122520), CD150-APC (Biolegend #115910), and CD48$^-$PE (Fisher #557485). Lin-Sca$^+$Kit$^+$ cells (LSKs) were sorted using a BD FACS Aria II cell sorter instrument (November 2008 edition) and BD FACSDiva software (version 8.0.1 2014). Sorted LSKs were grown overnight in SFEM medium containing 10 ng/ml murine IL-3, 10 ng/ml IL-6, 50 ng/ml SCF, 10 ng/ml thrombopoietin, and 20 ng/ml FLT3l. Cells were spinoculated with retrovirus expressing MSI2-ADA, MSI2-DCD, or MIG controls and 4 μg/mL polybrene on retronectin-coated plates. After 48 h of transduction, all cells were collected and transplanted into lethally irradiated C57BL/6 mice (15,000 cells per mouse). Engraftment was checked after 6 weeks. After 7 weeks of transplantation, mice were sacrificed and c-Kit enriched bone marrow cells were stained with LSK markers as described above plus CD48-PE and CD150-APC. Cells were sorted into four populations GFP positive CD150$^+$ CD48$^-$(LT-HSC), CD150$^-$ CD48$^-$ (ST-HSC or MPP1), CD150$^+$ CD48$^+$ (MPP2), and CD150$^-$ CD48$^+$ (MPP4). 360–20,000 sorted cells were used for RNA extraction and sequencing.

**MSI2-HyperTRIBE in LSKs and LSCs**. LSK cells were obtained and transduced with MSI2-HyperTRIBE constructs as described above. After 48 h of incubation, cells were sorted for GFP positive and RNA was extracted for SMARTer library preparation and RNA-seq.

Quaternary MLL-AF9 leukemia model on Actin-dsRed background mice were generated as described before[54]. Bone marrow cells were infected with MSI2-HyperTRIBE expressing virus in BMT medium (RPMI 10%FBS 1%L-Glutamine PenStrep supplemented with 10 ng/mL murine IL-3, 10 ng/mL murine IL-6, 10 ng/mL murine SCF, and 10 ng/mL murine GM-CSF) for 48 h. LSC-enriched population was isolated by sorting dsRed$^+$, GFP$^+$, and c-Kit-APC-Cy7 high (top 10–12%) for library preparation and RNA-seq.

**RNA extraction and sequencing**. RNA from cells suspended in Trizol was extracted with chloroform. Isopropanol and linear acrylamide were added, and the RNA was precipitated with 75% ethanol. Samples were resuspended in RNase-free water. For HyperTRIBE in MOLM-13, after PicoGreen quantification and quality control by Agilent BioAnalyzer, 1 μg RNA input was used for library preparation (TrueSeq Stranded mRNA LT Sample Prep Kit. Libraries were run on a HiSeq 4000 in a 50 bp/50 bp paired end run, using the HiSeq 3000/4000 SBS Kit (Illumina). The average number of read pairs per sample was 34 million. For HyperTRIBE in HSPCs, after RiboGreen quantification and quality control by Agilent BioAnalyzer, 0.5 ng total RNA (for eight samples with <0.5 ng, all mass was used) with RNA integrity numbers ranging from 1 to 9.9 underwent amplification using the SMART-Seq v4 Ultra Low Input RNA Kit (Clonetech catalog # 63488), with 12 cycles of amplification. Subsequently, 1–2 ng of amplified cDNA was used to prepare libraries with the KAPA Hyper Prep Kit (Kapa Biosystems KK8504) using eight cycles of PCR. Samples were barcoded and run on a HiSeq 4000 in a 50 bp/50 bp paired end run, using the HiSeq 3000/4000 SBS Kit (Illumina). An average of 40 million paired reads were generated per sample and the percent of mRNA bases per sample ranged from 69 to 82%. For HyperTRIBE in LSKs and LSCs, after RiboGreen quantification and quality control by Agilent BioAnalyzer, 2 ng total RNA with RNA integrity numbers ranging from 9.3 to 10 underwent amplification using

the SMART-Seq v4 Ultra Low Input RNA Kit (Clonetech catalog # 63488), with 12 cycles of amplification. Subsequently, 10 ng of amplified cDNA was used to prepare libraries with the KAPA Hyper Prep Kit (Kapa Biosystems KK8504) using eight cycles of PCR. Samples were barcoded and run on a HiSeq 4000 or HiSeq 2500 in High Output mode in a 50 bp/50 bp paired end run, using the HiSeq 3000/4000 SBS Kit or TruSeq SBS Kit v4 (Illumina). An average of 36 million paired reads were generated per sample and the percent of mRNA bases per sample ranged from 64 to 77%.

**Identification of RNA editing events in RNA-Seq data**. We aligned the paired-end RNA-seq reads to human (hg19) or mouse (mm10) genome using STAR aligner[55]. Next we followed the GATK[56] workflow for calling variants in RNA-seq (https://software.broadinstitute.org/gatk/documentation/article?id=3891) to identify all the mutations in each RNA-seq library. We then restricted to the mutations within annotated mRNA transcripts, as well as restricting to A-to-G mutations in transcripts encoded by the forward strand and T-to-C mutations in transcripts encoded by the reverse strand. We also filtered out mutations found in the dbSNP database since they are most likely DNA-level mutations. We then combined the filtered sets of RNA editing events from all RNA-seq libraries of the same experiment and counted the number of reads containing reference (A/T) and alternative (G/C) alleles from each library at each site.

**Statistical test for difference in edit frequencies**. We used beta-binomial distribution to model the RNA edit frequencies, which has also previously been applied to modeling allele frequencies in RNA-seq reads[57,58]. The beta-binomial distribution is the binomial distribution where the probability of success at each trial is not fixed, but instead is drawn from the beta distribution. The probability functions of the binomial distribution and beta distribution are:

$$P(k|n, p) = \binom{n}{k} p^k (1-p)^{n-k}, \tag{1}$$

$$\pi(p|\alpha, \beta) = \frac{p^{\alpha-1}(1-p)^{\beta-1}}{B(\alpha, \beta)}. \tag{2}$$

Thus the probability density function of the compound distribution, the beta-binomial distribution, can be represented as

$$
\begin{aligned}
f(k|n, \alpha, \beta) &= \int_0^1 P(k|n, p)\pi(p|\alpha, \beta)dp \\
&= \int_0^1 \binom{n}{k} p^k (1-p)^{n-k} \frac{p^{\alpha-1}(1-p)^{\beta-1}}{B(\alpha, \beta)} dp \\
&= \frac{\binom{n}{k}}{B(\alpha, \beta)} \int_0^1 p^{k+\alpha-1}(1-p)^{n+\beta-k-1} dp = \binom{n}{k} \frac{B(k+\alpha, n+\beta-k)}{B(\alpha, \beta)}.
\end{aligned}
\tag{3}
$$

For convenience, it is common to reparametrize it as:

$$\mu = \frac{\alpha}{\alpha+\beta}, \tag{4}$$

$$\rho = \frac{1}{\alpha+\beta+1}, \tag{5}$$

so that the expectation and variance of the beta-binomial distribution are:

$$E(k|n, \mu, \rho) = n\mu, \tag{6}$$

$$Var(k|n, \mu, \rho) = n\mu(1-\mu)[1 + (n-1)\rho]. \tag{7}$$

In this form, $\mu$ corresponds to the estimate of $p$, and $\rho$ corresponds to the extent of over-dispersion. Both $\mu$ and $\rho$ values are between 0 and 1.

When we use beta-binomial distribution to model the RNA editing events in RNA-seq, $n$ corresponds to the total number of reads overlapping with an RNA edit site and $k$ to the number of reads with A-to-G mutations. In this scenario, the beta-binomial distribution is a better model for read counts than the binomial distribution since it takes the variability in mutation frequencies between biological samples into account. Under the null hypothesis, all samples have equal RNA editing level, and the edit frequencies are drawn from the same beta distribution $\pi(\mu_0, \rho)$. Under the alternative hypothesis, the samples expressing the MSI2-ADA fusion protein have a different RNA edit frequency than the control samples, and the frequencies come from two different beta distributions $\pi(\mu_1, \rho)$ and $\pi(\mu_2, \rho)$. Using the read counts at each RNA edit site from biological replicates, we maximized the likelihood for both the null and alternative hypotheses and then computed the $p$ value using a likelihood ratio test. The $p$ values from all sites were adjusted to control for false discovery rate (FDR) using a Benjamin–Hochberg correction. The statistical computation was performed using R packages *VGAM* (Version 1.1–2) and *bbmle* (Version 1.0.23.1). Significant sites were determined by filtering for FDR-adjusted $p$ values, using FDR < 0.05 for MOLM-13, FDR < 0.01 for LSCs and LSKs and FDR < 0.1 for HSPCs. A target gene is retained if it has an expression level of at least 5 fpkm and at least one edit site with a significant differential edit frequency of at least 0.1 (differential edit frequency is the difference

in mean edit frequency by MSI2-ADA and mean edit frequency by MSI2-DCD and MIG).

**Statistical test for differential editing between cell types**. For differential editing between HSPC populations, we first identified all significantly edited genes with a maximum diff.frequency ≥ 0.1. A gene with a maximum diff.frequency ≥ 0.1 that is significantly edited in one cell type (ADAR vs controls), but not significantly edited in the other cell types (ADAR vs controls), is considered a potential cell-type specific gene target. Next, we obtained the read counts from all samples (LT, ST, MPP2, MPP4) supporting every A to G and T to C edit site and tested the significance for cell-type specific edit sites using the beta-binomial test. Under the null hypothesis, all cell types have equal RNA editing level, and the edit frequencies are drawn from the same beta distribution. Under the alternative hypothesis, the cell type of interest has a different RNA editing level than the other cell types. The difference in edit frequency between cell types is significant if the FDR-adjusted $p <$ 0.1. For the difference in editing between LSC and LSK-specific gene targets, we selected genes with a diff.frequency ≥ 0.6 and fpkm ≥ 5. These gene targets were run through the beta-binomial test as described above.

**Clustering of target genes by edit frequency patterns**. After identifying HSPC cell-type specific gene targets using the beta-binomial test, we filtered for adjusted $p < 0.1$ and plotted the maximum diff.frequency value for each gene. The diff. frequencies were then stacked from lowest to highest diff.frequency in each cell type.

After identifying genes significantly edited between LSCs and LSKs through the beta-binomial test, genes were filtered by an adjusted $p < 0.05$ and fpkm ≥ 5. We obtained the maximum diff.frequency (ADAR vs MIG/DCD) for each gene that passed the filter and plotted them in a heatmap with Mcquitty clustering method. GE heatmaps for both HSPCs and for LSKs and LSCs were created by using DESeq2 to obtain variance stabilized transformation (VST) of read counts. Then, we calculated the mean of the VST counts of sample duplicates/triplicates for each gene, and then performed z-transformation for each gene. Genes in the expression heatmap match the order of row in the edit frequency heatmaps.

**Motif analysis**. For de novo motif discovery, we first extracted sequences extending 100 bp from both sides of each edit site in the 3′UTR and considered all these windows as the target sequence pool for the HOMER program. Overlapping sequences were merged into a single sequence. Background sequences with length 201 bp were randomly selected from 3′UTRs in the genome that did not overlap with the target sequence pool. We used the HOMER software to search for enriched motifs of length 6, 7, or 8, and regional oligomer autonormalization of up to length 3.

To calculate the distance between the MSI2-HyperTRIBE edited site to the nearest MSI2 motif, we first obtained the genomic coordinates of exons that contain the HyperTRIBE site. Then we calculated the position weight matrix (PWM) of HOMER motif results to identify motif sites within exon sequences. A site was designated as a motif occurrence if its score was at least 90% of the maximum score; this score was calculated as the log of the probability of observing the nucleotide sequence given the motif PWM, divided by the probability of observing the given sequence at random given the background distribution of nucleotides, with a sampling correction applied to avoid null values[59]. We then calculated the distance of each edited site to the nearest motif match.

To find the distance to the nearest iCLIP peak, we then identified the genomic coordinate of the iCLIP peak nearest to each MSI2-HyperTRIBE edit site in MOLM-13 cells. NB4 iCLIP data from[21].

**MSI2 edit site clustering analysis**. To determine a suitable window size for clustering edit sites, we compared the enrichment of MSI2 motifs in windows of fixed size around significantly edited sites ("true sites") compared with windows of the same size around non-significantly edited sites ("background"). We performed a Fisher's test and determined that ±17 bp is the largest window such that the motif enrichment was significantly greater around true sites compared with background ($p < 0.01$).

**Differential expression analysis (DESeq2)**. Paired-end RNA-seq reads were first processed with Trimmomatic[60] to remove TruSeq adapter sequences and bases with quality scores below 20, and reads with <30 remaining bases were discarded. Trimmed reads were then aligned to mm9 genome with the STAR spliced-read aligner[55]. For each gene from the RefSeq annotations, the number of uniquely mapped reads overlapping with the exons was counted with HTSeq (http://www-huber.embl.de/users/anders/HTSeq/). Read counts were filtered by keeping all genes with a median read count ≥ 1 or mean rpkm or fpkm ≥ 1 and then used as input for DESeq2 to evaluate the difference in read counts of MOLM-13, different mouse HSPC populations, LSKs and LSCs expressing MSI2-DCD and those expressing MIG control. For differential expression of targets in LSCs and LSKs, only genes with fpkm ≥ 5 and edit frequency ≥0.1 were considered. A one-sided Wilcoxon test was performed to determine the statistical significance between the log2 fold changes (log2FC) of LSC unique, LSK unique, and shared targets.

**Gene pathway enrichment analysis**. Target genes in four populations of HSPCs were overlapped to identify the common and unique targets between the populations. Target genes specific for LT and ST HSCs or specific for MPP2 and MPP4 were analyzed for RNA-seq Gene and Drug signatures and Gene Ontology (molecular functions and biological processes) using ENRICHR program[34,61]. The same analysis was also done for targets unique to each population. The ENRICHR combined score was extracted for significantly enriched pathways and compared between different sets of targets. For pathway enrichment of GE independent targets, we first are defined GE independent targets as following. For shared and LSC unique groups, these are genes that have no significant expression difference between cell types (FDR ≥ 0.05) or comparable or lower expression in LSCs versus LSKs (log2FC LSC/LSK ≤ 0.26, equivalent to fold change LSC/LSK ≤ 1.2, and FDR < 0.05). For LSK unique group, GE independent targets are genes with no significant expression difference between cell types (FDR ≥ 0.05) or comparable or lower expression in LSKs versus LSCs (log2FC LSK/LSC ≤ −0.26, equivalent to fold change LSK/LSC ≤ 1.2, and FDR < 0.05).

**Immunofluorescence**. HSCs and MPPs were sorted from primary *Msi2* f/f Cre- and Cre+ 6 weeks after pIpC. Cells were fixed with 1.5% paraformaldehyde, permeabilized with cold methanol and cytospun onto glass slides. Cells were then stained on slides with anti-SMAD3 (Cell Signaling Technology, 9523S, dilution 1:1000), anti-phosphorylated SMAD2/3 (Cell Signaling Technology, 8685S, dilution 1:1000), or anti-BRCC3 (Novus Biologicals, NBP1-76831, dilution 1:1000) first and then with secondary antibody conjugated with rabbit Alexa Fluor 488 (Molecular Probes). Quantification of the signal intensity of each cells (divided by surface area) normalized for background staining was done with AxioVision Rel.4.8.2 (06-2010) software and Zeiss Imager Z2 (Zen 2 Blue Edition).

**Luciferase reporter assay**. Original or mutated 3′UTR of murine Hoxa9 and murine *c-Myb* was cloned downstream of Renilla luciferase reporter gene in pRL-CMV. MSI2 motifs in proximity of identified edit sites on *Hoxa9* and *Myb* 3′UTRs were located by "distance to nearest motif" R script, as described above, in LSKs and LSCs. All the motifs in *Hoxa9* and *Myb* 3′UTR were mutated. In the knock-down experiment, pRL-CMV 3′UTR constructs were co-transfected with firefly luciferase control and MSI2 shRNA or nonspecific shRNA control (shRNA scr). In the overexpression experiment, pRL-CMV 3′UTR constructs were co-transfected with firefly luciferase control and MIB empty vector or vector overexpressing human MSI2. After 48 h of transfection, expression of renilla and firefly luciferase was determined by Dual luciferase assay (Promega) following the manufacturer instructions.

**qRT-PCR**. Total RNA from sorted cKit-hi MLL-AF9 Msi2 RosaCre ER ± Tamoxifen cells was isolated using TRIzol (Sigma-Aldrich) and RNAeasy RNA extraction kit (Qiagen). RNA was reversed transcribed into cDNA with iScript (BioRad). Quantitative PCR was performed with primers for *Msi2* (forward ACGACTCCCA GCACGACC; reverse GCCAGCTCAGTCCACCGATA), *Ikzf2* (forward: CATCAC TCTGCATTTCCAGC; reverse: TGACCTCACCTCAAGCACAC), *Myb* (forward: AGATGAAGACAATGTCCTCAAAGCC; reverse: CATGACCAGAGTTCGAGC TGAGAA), and *Hoxa9* (forward: GTAAGGGCATCGCTTCTTCC; reverse: ACA ATGCCGAGAATGAGAGC).

**Immunoblot analysis**. To check the expression of *Hoxa9*, *Ikzf2*, and *Myb* in LSCs, c-Kit[hi] (top 10–12%) bone marrow cells (LSCs) from *Msi2* f/f Cre-ER- and *Msi2* f/f Cre-ER+ mice were sorted and were left untreated or treated with 600 nM 4-OH Tamoxifen (Sigma-Aldrich) for 68 h in BMT medium. One hundred thousand cells were collected, washed once with PBS, and then lysed in 1× Laemmli sample buffer (BioRad). LSCs were also sorted from quaternary MLL-AF9 DsRed leukemia mice, then were transduced with lentiviral shRNAs against murine *Msi2* (sh331 and sh332) or shRNA against Luciferase. Transduced cells were selected with 2 μg/mL puromycin. After 72 h of transduction, cells were collected, washed in PBS and lysed in 1× Laemmli sample buffer. For analysis in LSKs, one hundred thousand LSK cells from 3 week pIpC treated *Msi2* f/f Cre- and *Msi2* f/f Cre+ mice were sorted, washed with PBS and lysed in 1× Laemmli sample buffer. Cell lysate was run on 4–15% SDS-PAGE gels, transferred onto nitrocellulose membrane and then probed with antibodies against MSI2 (Abcam, ab76148, dilution 1:1000), HOXA9 (Abcam, ab140631; dilution 1:1000), IKZF2 (Santa Cruz, sc-9864, dilution 1:1000), MYB (Millipore, 05-175, dilution 1:1000), and ACTB (beta-actin-HRP, dilution 1:30,000) (Sigma-Aldrich, A3854).

**Reporting summary**. Further information on research design is available in the Nature Research Reporting Summary linked to this article.

## Data availability

All the RNA-seq data generated in this study have been deposited in the Gene Expression Omnibus database under the accession number GSE132949. The Msi2 knockdown in four human AML cell line microarray data referenced during the study are available in under accession number GSE22778. The Msi2 knockout in mouse LSKs microarray data referenced during the study are available in under accession number GSE53385. The

source data underlying all Figures and Supplementary Figures are provided as a Source Data file. A reporting summary for this article is available as a Supplementary Information file.

## Code availability

Custom codes used in this study are available at https://github.com/DiuTTNguyen/MSI2_HyperTRIBE_codes.

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

## Acknowledgements

We would like to thank members of the Kharas laboratory for their discussions, helpful advice, and suggestions. We would also like to thank the MSKCC Integrated Genomics Operation (IGO), Molecular Cytogenetics Core for their technical support. M.G.K. is a Scholar of the Leukemia and Lymphoma Society and supported by the US NIH National Institute of Diabetes Digestive and Kidney Diseases Career Development Award; NIDDK NIH R01-DK101989-01A1; NCI 1R01CA193842-01, R01HL135564, R01CA225231-01, and R01C186702-06; the Starr Cancer Consortium; the Alex's Lemonade Stand A Award; NYSTEM; the Susan and Peter Solomon Fund; and the Tri-Institutional Stem Cell Initiative 2016-014. C.P. is supported by NIDDK Research Supplement to Promote Diversity in Health-Related Research 3R01DK101989-03S1. L.P.V. is supported by K99 CA229993 and the LLS Career Development Award. The studies were supported by the MSK Cancer Center Support Grant/Core Grant (P30 CA008748). We would also like to thank Weijin Xu in Michael Rosbash laboratory at Brandeis University for his advice on the RNA extraction and sequencing methods. M.G.K. is a consultant for Accent Therapeutics and M.G.K.'s laboratory receives some financial support from 28-7. These disclosures are not directly related to these studies.

## Author contributions

D.T.T.N. led this project, designed and performed experiments, analyzed data, and wrote the manuscript. Y.L. developed the computational pipeline, and Y.L. and E.L.C. both performed bioinformatic analysis and wrote the manuscript. X.Y. performed experiments. S.M.P. performed experiments and provided critical suggestions. Z.C. and C.C. performed bioinformatic analysis. C.P., A.S., E.S., A.M.S., S.G., and P.P. all performed experiments. L.P.V. provided critical suggestions. C.L. provided critical suggestions, supervised the project, and edited the manuscript. M.G.K. directed the project, analyzed data, and wrote the manuscript.

## Competing interests

The authors declare no competing interests.
