## [Peer Review File · Nature Communications]

Reviewers' comments:

Reviewer #1 (Remarks to the Author):

Nguyen and colleagues use a relatively novel technique – HyperTRIBE (used for the first time in mammalian cells) to characterize the RNA interactome of the RNA binding protein MSI2. Given the low cell requirement of HyperTRIBE the authors are able to evaluate the binding of MSI2 in rare cell populations, such as LT-HSC and leukemia stem cells (LSCs). The paper is potentially interesting but might require some clarifications before it is suitable for publication.

Major points:

Fig.1 – What is the effect of overexpression of a construct with ADAR activity linked to a MSI2 protein lacking RNA binding? It seems like a better control to assess the aspecific ADAR activity due to overexpression.

Fig.2. What is the change in expression between MIG and MSI2-ADA? And MSI-ADA and MSI2-DCD? It would be good to know what are the changes in expression induced by MSI2 overexpression.

Fig.2 It is a bit disappointing that out of all the new hits and targets, the authors focused on a target (SMAD3) which was previously picked up also using other techniques. It would be great to showcase a novel target, specific for the LT population, that has been uncovered by HyperTRIBE (due to the requirement of low-input material). Same comment applies for Fig.4 and the LSCs targets.

Fig.3. It is not clear if the comparison in differential expression is between LSC and LSK that are untransfected? Or those overexpressing MSI2? Is there an effect in transcription upon MSI2-ADA overexpression?

Fig.4. What are the levels of IKZF2, MYB, MYC and HOXA9 across cell types (LSKs and LSCs)? It would be great to compare the levels in the same blots, to evaluate the effect of increased MSI2 binding in LSCs.

Minor points:

- References seem to be repeated twice in the text.
- It would be good to cite "Rahman et al Nature Protocols 2018"
- Fig.S2. A similar IF for P-SMAD3 should be shown
- if my understanding of the final message is correct, I would expect higher levels of MSI2 targets in LSCs. This could be better represented in the final model (panel 4F).

Reviewer #2 (Remarks to the Author):

RNA-binding proteins are crucial for proper development, and understanding their targets and changes in their activity is key to elucidating many developmental processes. The manuscript of Nguyen et al. describes adaptation of a newly developed technique called HyperTRIBE to a mammalian system, and application of this technique to human and mouse stem and progenitor cells as well as leukaemia stem cells. In particular, the authors study the RNA-binding protein MSI2 and its changing RNA targets, detected by changes in A-to-G editing near where the protein binds in this protocol. Studying protein-RNA interactions in rare cell types and across dynamic biological processes remains a challenge, and I read the manuscript with great interest.

I have a number of concerns I would like the authors to address before recommending this manuscript for publication in Nature Communications. These relate broadly to the implied improvements over existing alternative approaches, novelty, definition of edit frequency, the possible impact of endogenous RNA editing and over-expression of the MSI-ADA construct, and the overall functional impact of MSI2 binding on target RNAs. These and a number of other points are described in more detail below.

MAJOR POINTS:

(1) Technical improvements over existing alternative approaches:

The manuscript would benefit from a more in-depth experimentation and discussion on sensitivity of the approach. Based on the abstract and introduction (page 1, lines 19-23; page 2, lines 51-55), I expected the authors to present and discuss a method that can be used when the material (cell numbers) are limited. However, the manuscript does discuss the rarity of the cell material they are working with or give numbers on how many cells the protocol requires and how this compares with other protocols. Therefore, it remains unclear from the current manuscript whether HyperTRIBE truly performs well on rare cell material and does so better than existing alternatives.

Overall, the manuscript would benefit from a detailed comparison between their new approach and the existing alternatives that have already been used to study MSI2 binding. In what situations would one want to use HyperTRIBE rather than other CLIP-based approaches, and vice versa?

(2) Biological novelty:

The binding-site data presented in this manuscript is valuable as it replicates and adds to MSI2 binding sites detected previously with alternative approaches also in other cell lines. However, as MSI2 binding has been studied before using similar high-throughput approaches, I am not convinced that there is enough novelty in this study in its current form in terms of method and MSI2 binding sites and mechanisms to warrant publication in Nature Communications. I found the experiments and analyses on long term, short term, and multipotent progenitor cells particularly interesting – could the authors expand on this work with a few more targeted experiments and analyses to relate MSI2 binding functional differences in these cell types?

(3) Calculation of edit frequency:

This analysis is non-trivial and impacts substantially any conclusions drawn from these data. Therefore, I would like to see more details on how significantly edited (bound) sites are calculated, and graphs on to justify the chosen approach and thresholds, in the main text. How is replicate information taken into account? How dependent is binding site detection on sequencing depth? For example, it is surprising that so many sites with edit frequency of < 0.1 can be identified as significant (Fig 1C). Is this for genes with very high expression? Is there a need to use threshold on minimum number of reads? 1 edit in total 10 reads gives freq of 0.1 but this can presumably happen quite easily by chance.

How do the authors define significantly different editing between cell types? Frequency of edit sites fluctuates more easily for sites that are not detected by many reads than those that have tens or hundreds of reads. Does a change from edit frequency of 0.4 to 0.6 mean a relevant change in binding? A number of representative examples of individual genes or binding sites showing a pile-up of edited and total (non-edited) reads in controls and test cell lines would be useful. Similarly for edit sites with significant differences between cell lines.

Do the authors cluster nearby edit sites in any way? ADAR linked to MSI2 via a flexible linker will presumably edit any suitable adenosine near the binding site, which may inflate the final number of binding sites. Is there a need to group edit sites in near proximity to formulate binding regions? Please discuss, and re-analyse if appropriate.

(4) Control for endogenous RNA editing and ADAR expression:

With a method that relies on detection of editing on RNA, it is critical to be able to account for background that comes from endogenous RNA editing activities and sequencing errors. The authors use a catalytically dead version of MSI-ADA fusion as an appropriate control. I would like the authors to present more data on how these controls were analysed and interpreted, as this is not clear from the current manuscript. I would also like to see an experiment to show that endogenous ADAR activity is not different between the various cell lines the authors compare to rule out its impact on the observed differences in A-to-I editing.

(5) Impact of over-expression of the MSI-ADA construct:

Do the authors think that over-expression of MSI-ADA construct might lead to MSI binding to sites it normally would not interact with? Is there a way to try to distinguish 'true' binding events from such artificial sites, for example by setting stringent criteria on replication and a high(er) threshold on edit frequency and total number of reads to support any give edit site? How sensitive is the method to possible differences in expression of the MSI-ADA construct when comparing different cell types? Please discuss potential pitfalls of MSI-ADA overexpression and any solutions.

(6) Categorisation of target (edited) RNAs by cell type:

I find the visualisation and grouping based on editing frequencies in Figures 2E and 3F misleading. Is there a biological or technical basis for highlighting a specific range in editing over others? For example, is there a biological or technical reason to say that a change from edit frequency 0.4 to 0.6 is much more indicative of a drastic change in binding than a change from 0.2 to 0.4 or from 0.6 to 0.8? This is what Figure 3F conveys visually due to the colour scale chosen to display edit frequencies. This visual threshold is also set differently in the two figures: In Figure 2F the chosen colour scheme emphasises differences in 0-0.2 range over 0.2-0.8, and in Figure 3F it emphasises differences in 0.4-0.6 range over other similarly sized ranges. Based on the evidence provided, I am not convinced that the grouping of genes into the presented clusters is meaningful. Please display this data on a continuous colour scale that gives a truthful visual impression of differences in edit frequencies, and discuss if there is a way to assess what level of changes in editing are indicative of true differences in binding.

(7) Functional impact of MSI2 binding:

The manuscript does not go very far in explaining the functional impact on MSI2 binding at a molecular level and on cell-type differences. It is interesting that MSI2 binding does not seem to have a connection with RNA expression levels (Figs 2E and 3F), but this begs the question – what impact does MSI2 binding have? The authors confirm that MSI2 depletion impact protein levels of two previously known target genes and one novel one, suggesting a role for MSI2 in translation. I would hope to see a more proteome-wide analysis of the impact of MSI2 binding to strengthen this point in a journal like Nature Communications. I would also like to see an experiment that directly shows that MSI2 binding affects translation (e.g. by mutating the binding site and observing target RNA & protein abundance).

(8) Impact of artificial editing on the transcriptome:

A-to-I editing exist in cells to regulate RNA life cycle. Is there a risk that high expression of exogenous ADAR, used here to identify binding regions, artificially influences the expression and processing of those RNAs? Please discuss in text.

INTERMEDIATE POINTS:

(1) Previous research on RNA-binding proteins in context of stem cell:

Page 1 lines 16-18: "However, the aspect of cell-context dependent activity of RBPs during normal stem cell differentiation and transformation to malignancies has never been addressed" and page 2 lines 43-43: "However, whether RBPs may have cell-type specific activity between different cellular states of normal stem cell differentiation or between normal and transformed cells has never been addressed". A good number of studies have looked at RNA-binding proteins in stem cells, differentiated cells, and malignant cells, although many of these studies may not have taken a high-throughput approach. I agree with the authors with the importance of studying the activity of RBPs in a cell-type specific manner during cell differentiation, but I would recommend they tone down this statement and discuss and cite the work done in this realm. This could include, for example, some the following original research and review articles:

- Li and Izpisua Belmonte. Deconstructing the pluripotency gene regulatory network. *Nat Cell Biol.* 2018. PMID: 29593328
- Yang et al. Imp and Syp RNA-binding proteins govern decommissioning of Drosophila neural stem cells. *Development.* 2017 PMID: 28851709
- Hayakawa-Yano et al. An RNA-binding protein, Qki5, regulates embryonic neural stem cells through pre-mRNA processing in cell adhesion signaling. *Genes Dev.* 2017. PMID: 29021239
- Ju Lee et al. A post-transcriptional program coordinated by CSDE1 prevents intrinsic neural differentiation of human embryonic stem cells. *Nat Commun.* 2017. PMID: 29129916
- Degrauwe et al. IMPs: an RNA-binding protein family that provides a link between stem cell maintenance in normal development and cancer. *Genes Dev.* 2016. PMID: 27940961
- Wurth and Gebauer. RNA-binding proteins, multifaceted translational regulators in cancer. *Biochim Biophys Acta.* 2015. PMID: 25316157
- Kwon et al. The RNA-binding protein repertoire of embryonic stem cells. *Nat Struct Mol Biol.* 2013. PMID: 23912277

(2) 'Gene and Drug' enrichment analyses:

Figures 2F, 3H-K: Please elaborate on how the listed significant Gene&Drug signatures indicate stemness or differentiation, and what the stemness/differentiation gradient in Figure 2F is based on. The terms seem to related to specific knock-downs and knock-outs in different cell types, for example to previously done MSI2 experiments. This is perhaps technically encouraging, but I am wondering if this analysis tells more about shared RNA targets than stemness/differentiation per se. Admittedly, I am not familiar with the Gene&Drug signatures resource, so perhaps an expanded description of the resource and method would go a long way here. Would a Gene Ontology (GO) enrichment analysis give a more straight-forward indication of whether the targets are involved in stemness and differentiation? Could the authors try this?

(3) Technical details in text:

Please include more technical details in the main manuscript. For example, please state clearly in the text how many replicates were analysed for each experiment. Minimum of 2 is required for

RNA-seq based experiments, ideally 3 (or more).

(4) Details on data availability:

Please indicate clearly the database and IDs where the raw and processed data will be publicly available upon publication (for example, Gene Expression Omnibus or ArrayExpress).

MINOR POINTS:

(1) Figure 1 panel D:

Does this graph take into account the relative length of these features? 3'UTRs are probably considerably longer than CDS and 5UTR regions for a large fraction of genes. To show preference for 3'UTRs, one would ideally take their relative length into account.

(2) Figure 1 panel E:

Why is the MSI1 included in the label?

(3) Figure 1 panel F:

I am unclear on whether the distance is calculated from the motif (motif-to-hyperTRIBE site and motif-to-iCLIP site) or the HyperTRIBE binding site (hyperTRIBE-to-motif and hyperTRIBE-to-iCLIP). Please clarify. Either way, please make clear that that this plot does not indicate that iCLIP identifies the binding sites less precisely than HyperTRIBE.

(4) Figure 1 panel G:

Could the authors please clarify what is plotted here and how to interpret the graph? Are the 255 genes on the x-axis? The y-axis seems to be cut off at the bottom. What does 'NES' stand for? Does the FDR cutoff refer to the hyperTRIBE done in this manuscript, or the dataset it is being compared to (Kharas et al. 2010)? What does the bottom part of the graph represent (black barcode), and what does the red and blue scale mean? I am also confused about the placement of the labels 'Ctrl', 'MSI2 knock-down UP' and 'MSI2 knock-down'.

(5) Figure 3 panel J legend:

The description "Gene expression (GE) independent RNA-seq Gene and Drug signatures" is a little confusing to me. Could it be clarified and simplified to "Significantly enriched Gene and Drug signatures based on ENRICH analysis of shared MSI2 targets in LSKs and LSCs"?

(6) Figure 3 panel K legend:

Similarly to legend J, I find the description "GE independent signature of RNA-seq" unclear. Please modify.

(7) Figure 4 panels A-C:

Could the authors show these binding sites with respect to their real position along the (exonic regions) 3'UTR rather than in an abstracted order? Something like a nice version of a genome-browser screenshot. It would be informative for the reader to see if and how the binding sites cluster along the 3'UTR. It would also be very informative to see in this plot, perhaps on a separate track, the edit frequency in control cells, as well as to total number of reads the frequency calculation is based on. Does the significance test and 'ns' label refer to comparison between LSK and LSC at a given site? Please clarify in the legend.

(8) Sharing the code so that the scientific community can build upon it:

I would like to encourage the authors to make all the code written to analyse these data available, for example on GitHub or as a supplementary to the manuscript. As this may not be Nature Communications policy, I do not consider it necessary for publishing this manuscript. The authors mention the R script for distance-to-motif analysis is available on GitHub. Please provide the URL.

Reviewer #3 (Remarks to the Author):

The manuscript by Nguyen et al employs the HyperTRIBE approach to reveal MSI2 RNA binding network in haematopoietic stem cells (HSCs) and leukaemic stem cells (LSCs). Multiple RNA binding proteins have been found to play key regulatory roles in normal haematopoiesis and leukaemic transformation. However, small cell numbers have always been a limiting factor in identifying their target transcripts at high resolution. In this manuscript, the authors have optimised and used HyperTRIBE method, which so far has worked in Drosophila cells but has never been employed in mammalian cells. This paper not only provides a substantial technological advance but also reveals important insights into how MSI2 functions in normal and malignant haematopoiesis.

My comments and suggestions are listed below.

1) Given the novelty of this method in mammalian cells, it would be important that the authors include the full sequences of MSI2-DCD and MSI2-ADA for the benefit of the research community.

2) It is noted that human MSI2 was used. What was the rationale behind using human MSI2 if the goal was to perform the study in murine cells? Are mouse and human MSI2 proteins conserved?

3) Human AML cells were used to optimise the HyperTRIBE approach. This gives the opportunity to compare MSI2- interacting transcripts between human and mouse leukaemic cells at high resolution. What was the degree of an overlap between MSI2-binding transcripts between human and mouse cells? Are the targets conserved? What are they?

4) Regarding data presented in Figure 2 – these are very exciting results highlighting the feasibility of high-resolution editing in small numbers of cells. Do the authors have data indicating that HSCs were sorted from recipients with multilineage reconstitution?

5) The authors conclude that "Msi2 depletion was likely altering SMAD3 translation specifically in LT-HSCs compared to ST-HSC and MPPs (Park et al., 2014)." They go on to state: "These data indicate that despite similar abundance of MSI2 and its targets, MSI2 can differentially control its targets' expression during hematopoietic differentiation...". Does MSI2 control translation of other direct HSCs-specific targets identified by the HyperTRIBE approach?

6) In the discussion, that authors state: "Importantly, our results demonstrate that RBP-RNA interactions are highly cell- context dependent. We revealed that MSI2 has differential binding activity at different states of HSPCs and in LSCs in a target gene expression independent manner. It would be very informative to expand this section and offer some potential explanation for the cell-context dependency of MSI2 and the stronger editing in LSCs vs HSCs. Furthermore, why does MSI2 controls HOXA9, IKZF2 and MYC protein expression in LSCs but not HSCs? It would be desirable to discuss these data in more detail.

Minor points:

- 1) Figure 2A – schematic - should be "irradiation" not "irridiation".
- 1) For consistency, all genes and transcripts should be italicised in the figures.

Reviewer #1's comments; expert in RNA editing:

Nguyen and colleagues use a relatively novel technique – HyperTRIBES (used for the first time in mammalian cells) to characterize the RNA interactome of the RNA binding protein MSI2. Given the low cell requirement of HyperTRIBES the authors are able to evaluate the binding of MSI2 in rare cell populations, such as LT-HSC and leukemia stem cells (LSCs). The paper is potentially interesting but might require some clarifications before it is suitable for publication.

Major points:

Fig.1 – What is the effect of overexpression of a construct with ADAR activity linked to a MSI2 protein lacking RNA binding? It seems like a better control to assess the aspecific ADAR activity due to overexpression.

As suggested by the reviewer, we have carried out three additional HyperTRIBES experiments in MOLM13 cells using ADAR catalytic domain alone or ADAR fused to (1) MSI2 lacking both RRMs and (2) MSI2 with both RRM mutated at 5 amino acids important for RNA binding activity (F24A, R62A, F66A on RRM1; F223A and F155A on RRM2). As expected, loss of MSI2 or MSI2's RNA binding ability dramatically reduces ADAR editing targets, as these controls generate only a small set of significant editing sites (52 sites in ADA alone, 18 sites in RRM deleted and 20 sites in RRM mutated fusions with frequency above the background, respectively) compared to the MSI2-ADAR fusion (5244 sites). This suggests that the MSI2-ADAR editing activity reported in our HyperTRIBES is dependent on and reflects specific RNA binding activity of MSI2 to its targets. These data are now provided in Supplemental Figure 1 of the revised manuscript. The overlapping of edit sites is attached below for the reviewer's reference.

Overlapping significant edit sites by MSI2-ADA, RRM(del)MSI2-ADA, RRM(mut)MSI2-ADA and ADA only

Fig.2. What is the change in expression between MIG and MSI2-ADA? And MSI-ADA and MSI2-DCD? It would be good to know what are the changes in expression induced by MSI2 overexpression.

We have provided these additional analyses in Supplemental Figure 1 and Supplemental Figure 3. Our data show little or only modest changes in the gene expression of cells expressing MSI2-ADA or MSI2-DCD within 48hrs in MOLM13, LSK and LSC.

- MOLM13 MSI2-ADA vs MSI2-DCD: no significant differential expressed genes with FDR<0.05
- MOLM13 MSI2-ADA vs MIG: 10 significant differential expressed genes with FDR<0.05
- LSK MSI2-ADA vs MSI2-DCD: no significant differential expressed genes with FDR<0.05
- LSK MSI2-ADA vs MIG: no significant differential expressed genes with FDR<0.05
- LSC MSI2-ADA vs MSI2-DCD: no significant differential expressed genes with FDR<0.05
- LSC MSI2-ADA vs MIG: 8 significant differential expressed genes with FDR<0.05

Fig.2 It is a bit disappointing that out of all the new hits and targets, the authors focused on a target (SMAD3) which was previously picked up also using other techniques. It would be great to showcase a novel target, specific for the LT population, that has been uncovered by HyperTRIBE (due to the requirement of low-input material). Same comment applies for Fig.4 and the LSCs targets.

We now have used a stringent statistical test (See the Methods part about beta binomial test) to test the differential editing of MSI2-ADAR, which reflects differential binding of MSI2, at LT-HSC unique targets in LT-HSCs compared to in ST-HSCs and MPPs, as well as other groups of genes that are unique to other populations. We have provided the data in Figure 2E of the revised manuscript. Among these, *Brcc3* has been reported to be mutated in AML and MDS, and these mutations are associated with clonal hematopoiesis suggesting a function in HSCs (Huang et al., 2015; Meyer et al., 2019). Therefore, we selected this target to further validate by immunofluorescence. We found that BRCC3 protein level is significantly reduced in LT-HSC but not in ST-HSC and MPPs upon *Msi2* deletion. Our previous data shows no change in mRNA level, suggesting that *Brcc3* is a novel translational target of MSI2 in HSC.

Furthermore, although *Smad3* was previously identified, this study provides a new insight in that *Smad3* binding and regulation by MSI2 is differential in HSCs versus MPPs, which was not possible using other techniques.

In addition, we have identified *Myb* as a novel MSI2 target in LSCs, apart from the known targets *Hoxa9* and *Ikzf2*, which we now show to be differentially regulated in LSCs compared to LSKs.

Fig.3. It is not clear if the comparison in differential expression is between LSC and LSK that are untransfected? Or those overexpressing MSI2? Is there an effect in transcription upon MSI2-ADA overexpression?

The differential expression provided in Figure 3D is between LSC and LSK expressing empty vector MIG. Additional differential expression analysis between LSC and LSK expressing MSI2-ADA (attached here) results in similar conclusion that almost all shared and the majority

of LSC unique targets have comparable expression in both cell types or lower expression in LSKs whereas the majority of LSK specific targets were expressed more highly in LSKs. The differential expression in Figure 3F is between LSC and LSK expressing MSI2-ADA and empty vector (MIG) as indicated. This data is meant to demonstrate that the differential editing is not originating from differential expression from MSI2-ADA or MIG induction.

There is little effect in transcription (gene expression) upon MSI2-ADA overexpression in LSC and LSK within 48hr as shown in Supplemental Figure 3J-O.

Fig.4. What are the levels of IKZF2, MYB, MYC and HOXA9 across cell types (LSKs and LSCs)? It would be great to compare the levels in the same blots, to evaluate the effect of increased MSI2 binding in LSCs.

We have provided immunoblotting of these proteins on the same blot in Supplemental Figure 4F in the revised manuscript. Interestingly, we observe only a modest increased abundance of HOXA9, IKZF2 and MYB in LSC compared to LSK. However, as shown in Figure 4, all of these targets are reduced in LSC but not LSK upon MSI2 depletion. Therefore, these results suggest that MSI2 may not upregulate but is more required in LSC to maintain the expression of these important targets.

Minor points:

- References seem to be repeated twice in the text.

We have removed the repeated reference in the revised manuscript.

- It would be good to cite “Rahman et al Nature Protocols 2018”

We have cited this additional reference in the revised manuscript.

- Fig.S2. A similar IF for P-SMAD3 should be shown

We have provided IF images of P-SMAD3 in Supplemental Figure 2L

- if my understanding of the final message is correct, I would expect higher levels of MSI2 targets in LSCs. This could be better represented in the final model (panel 4F).

We have run an immunoblot to compare the indicated MSI2 targets in LSK and LSC (Supplemental Figure 4C). Interestingly, we observe only a modest increased abundance of HOXA9, IKZF2 and MYB in LSC compared to LSK. However, as shown in Figure 4, all of these targets are reduced in LSC but not LSK upon MSI2 depletion. These results suggest that LSC is more dependent on MSI2 to maintain the expression of these important targets.

Reviewer #2's comments; expert in RNA binding proteins:

RNA-binding proteins are crucial for proper development, and understanding their targets and changes in their activity is key to elucidating many developmental processes. The manuscript of Nguyen et al. describes adaptation of a newly developed technique called HyperTRIBE to a mammalian system, and application of this technique to human and mouse stem and progenitor cells as well as leukaemia stem cells. In particular, the authors study the RNA-binding protein MSI2 and its changing RNA targets, detected by changes in A-to-G editing near where the protein binds in this protocol. Studying protein-RNA interactions in rare cell types and across dynamic biological processes remains a challenge, and I read the manuscript with great interest.

I have a number of concerns I would like the authors to address before recommending this manuscript for publication in Nature Communications. These relate broadly to the implied improvements over existing alternative approaches, novelty, definition of edit frequency, the possible impact of endogenous RNA editing and over-expression of the MSI-ADA construct, and the overall functional impact of MSI2 binding on target RNAs. These and a number of other points are described in more detail below.

MAJOR POINTS:

(1) Technical improvements over existing alternative approaches:

The manuscript would benefit from a more in-depth experimentation and discussion on sensitivity of the approach.

Information for the methods for HyperTRIBE in each cell type used in the study can be found in our revised methods section. We also described the number of cells, the amount of total RNA used for RNA-seq, the library preparation method and sequencing depth for HyperTRIBE in each cell type.

Based on the abstract and introduction (page 1, lines 19-23; page 2, lines 51-55), I expected the authors to present and discuss a method that can be used when the material (cell numbers) are limited. However, the manuscript does discuss the rarity of the cell material they are working with or give numbers on how many cells the protocol requires and how this compares with other protocols. Therefore, it remains unclear from the current manuscript whether HyperTRIBE truly performs well on rare cell material and does so better than existing alternatives.

We indicated in the methods the number of cells required for HyperTRIBE in each cell type used for this study. The standard methods (RNA-IP and CLIPs) require typically 5-20 millions of cells (HITS-CLIP More M. et al., 2014; iCLIP Huppertz I. et al. 2014; eCLIP Nostrand E. et al, 2016; sCLIP Kargapolova Y. et al., 2017). ICLIP method for low input material requires 20,000-100,000 cells (Zarnegar B. et al., 2016). However, all of these CLIP methods require cross-

linking and RBP Immunoprecipitation (IP) which result in non-specificity and reduced recovery. Our HyperTRIBE method, as indicated in the Methods section, uses between 0.5 million cells (for MOLM13) to 360 cells (for LT-HSC) and does not need any cross-linking, IP or labeling steps. We show in all of the cell types used in our study that HyperTRIBE sites enriched for the relevant motif (MSI2), and we validated the targets by comparing with previous methods and genetic studies (depleting MSI2 and checking the target expression). We believe that this method for the first time allows for the measurement of RBP targets in rare cells types in as low as 360 cells as we tested. We did not test the method to its limit, which we considered to be out of scope for this work, but instead demonstrate that it can be utilized for standard experimental cell numbers that are currently not feasible with any other RBP IP-based mapping method.

Overall, the manuscript would benefit from a detailed comparison between their new approach and the existing alternatives that have already been used to study MSI2 binding.

We have not seen any published data on MSI2 binding in HSCs or LSCs using the existing alternatives; therefore, we are unable to make the requested comparison, which is another reason why this study is important. However, to validate the accuracy and sensitivity of HyperTRIBE in mammalian cells we have made the comparison of MSI2 binding identified by HyperTRIBE and by CLIP-based methods in cell lines. We show that HyperTRIBE is consistent with other methods in determining the MSI2 motif, that its binding sites are close to those determined by CLIPs, and that top hits in HyperTRIBE are also top binders in CLIPs.

In what situations would one want to use HyperTRIBE rather than other CLIP-based approaches, and vice versa?

As the reviewers rightly pointed out in a number of publications shown below, CLIP-based approaches have been successfully used to study RBP targets in embryonic stem cells since these cells can be isolated in a large number during embryonic development. However, to our knowledge there are no standard methods including CLIP-based techniques have been used to identify RBP targets in adult stem cells which appear in much lower frequency. For example, the frequency of HSCs in adult bone marrow is 0.01% of total nucleated cells and approximately 5000 can be isolated from an individual mouse. LSCs also represent a low-frequency subpopulation of leukemia cells. Therefore, our adapted HyperTRIBE method would benefit greatly studies about functions/targets of an RBP of interest in rare cells such as adult stem cells or patient samples. Additionally, this method can also be applied to identify RBP targets in mammalian cell lines as it is technically simpler and less laborious than any standard approaches: it requires no cross-linking, no immunoprecipitation and radioactive work (that is used in CLIP-based methods).

Besides the utility of being able to perform HyperTRIBE on rare cells it is also possible that CLIP has its own issues due to quality of antibodies or issues with pull down conditions. Various CLIP procedures (PAR-CLIP, iCLIP, HITS-CLIP) and RIPs pull down a range of non-specific targets, and it is critical to examine differential CLIP relative to control experiments to eliminate these (e.g. Loeb et al., *Molecular Cell* 2012). Although we find a strong overlap with a previous CLIP approach it is likely that HyperTRIBE depending on the RBP might pick up unique targets that might not be picked up by CLIP for various reasons. Overall, potentially multiple strategies

might be needed for a single RBP to obtain the full list or the most robust and reproducible targets.

(2) Biological novelty:

The binding-site data presented in this manuscript is valuable as it replicates and adds to MSI2 binding sites detected previously with alternative approaches also in other cell lines. However, as MSI2 binding has been studied before using similar high-throughput approaches, I am not convinced that there is enough novelty in this study in its current form in terms of method and MSI2 binding sites and mechanisms to warrant publication in Nature Communications.

Although MSI2 binding sites have previously been identified in cell lines using alternative approaches, MSI2 binding in HSPCs and LSCs has never been characterized. It was not clear before whether MSI2 binding sites in leukemia cell lines (NB4, K562) and other non-hematopoietic primary cells such as intestinal cells or cell lines (293T) are conserved to HSCs and LSCs. Using HyperTRIBE, we were able to discover MSI2 binding sites across the transcriptome in rare adult stem cells for the first time. Furthermore, using this method, we uncovered new biology that an RBP binding activity can be cell-context dependent. More specifically, we observed that the MSI2 targeting program changes when HSCs differentiate into more committed progenitors. We showed that MSI2 binding activity increases dramatically in leukemic stem cells compared to normal HSPCs. Previously it was understood that MSI2 is more required in LSCs than HSPCs. However, it was not clear whether this was due to higher expression of MSI2 in LSCs. In this study, we provide a new insight into this question, that the higher requirement of MSI2 in LSCs may result from a higher binding activity (with more RNA targets and more binding sites per target) despite lower or equivalent MSI2 expression in these cancer stem cells.

Most importantly, this is of broad relevance beyond interest in MSI2, as it opens up the question that RBP activity can be dynamically regulated during cell fate transitions or between different cell types even if they are expressing similar levels of that RBP. This also has clinical implications potentially suggesting that certain RBPs could be therapeutic targets without dysregulated expression but altered activity. Our study opens the possibility to explore these questions with many RBPs that are important in biology and disease.

I found the experiments and analyses on long term, short term, and multipotent progenitor cells particularly interesting – could the authors expand on this work with a few more targeted experiments and analyses to relate MSI2 binding functional differences in these cell types?

We now have used a stringent statistical test (see the Methods for a description of the beta binomial test) to test the differential edit frequency of MSI2-ADAR, which reflects differential binding of MSI2, at LT-HSC unique targets in LT-HSCs compared to in ST-HSCs and MPPs, as well as other groups of genes that are unique to other populations. We have provided the data in Figure 2E of the revised manuscript. Among these, *Brcc3* has been reported to be mutated in AML and MDS, and these mutations are associated with clonal hematopoiesis suggesting an important function in HSCs. Therefore, we selected this target to further validate by immunofluorescence. We found that BRCC3 protein level is significantly reduced in LT-HSC

but not in ST-HSC and MPPs upon *Msi2* deletion. Our previous data shows no change in mRNA level, suggesting that *Brcc3* is a novel translational target of MSI2 in HSC. Although *Smad3* was previously identified, this study provides a new insight in that *Smad3* binding and regulation by MSI2 is differential in HSCs versus MPPs, which was not possible using other techniques.

We agree with the reviewer that it would be very interesting to expand on the functional study on these targets in the subpopulations of HSPCs. However, such a study requires genetic approaches and transplantation in model animals and it is beyond the scope of this paper.

(3) Calculation of edit frequency:

This analysis is non-trivial and impacts substantially any conclusions drawn from these data. Therefore, I would like to see more details on how significantly edited (bound) sites are calculated, and graphs on to justify the chosen approach and thresholds, in the main text. How is replicate information taken into account? How dependent is binding site detection on sequencing depth? For example, it is surprising that so many sites with edit frequency of < 0.1 can be identified as significant (Fig 1C). Is this for genes with very high expression? Is there a need to use threshold on minimum number of reads? 1 edit in total 10 reads gives freq of 0.1 but this can presumably happen quite easily by chance.

We have now edited the methods and expanded them to include the reviewer's questions about this as we agree these are all important considerations. The methods section we provide is written below:

Identification of RNA editing events in RNA-Seq data

We aligned the paired-end RNA-seq reads to human (hg19) or mouse (mm10) genome using STAR aligner (Dobin et al., 2013). Next we followed the GATK (Van der Auwera et al., 2013) workflow for calling variants in RNA-seq (<https://software.broadinstitute.org/gatk/documentation/article?id=3891>) to identify all the mutations in each RNA-seq library. We then restricted to the mutations within annotated mRNA transcripts, as well as restricting to A-to-G mutations in transcripts encoded by the forward strand and T-to-C mutations in transcripts encoded by the reverse strand. We also filtered out mutations found in the dbSNP database since they are most likely DNA-level mutations. We then combined the filtered sets of RNA editing events from all RNA-seq libraries of the same experiment and counted the number of reads containing reference (A/T) and alternative (G/C) alleles from each library at each site.

Statistical test for difference in edit frequencies between conditions

We used beta-binomial distribution to model the RNA edit frequencies, which has also previously been applied to modeling allele frequencies in RNA-seq reads (Parker et al., 2016; Yablonovitch et al., 2017). The beta-binomial distribution is the binomial distribution where the probability of success at each trial is not fixed but instead is drawn from the beta distribution. The probability functions of the binomial distribution and beta distribution are:

$$P(k|n, p) = \binom{n}{k} p^k (1 - p)^{n-k}$$

$$\pi(p|\alpha, \beta) = \frac{p^{\alpha-1}(1-p)^{\beta-1}}{B(\alpha, \beta)}$$

Thus the probability density function of the compound distribution, the beta-binomial distribution, can be represented as

$$\begin{aligned} f(k|n, \alpha, \beta) &= \int_0^1 P(k|n, p) \pi(p|\alpha, \beta) dp \\ &= \int_0^1 \binom{n}{k} p^k (1-p)^{n-k} \frac{p^{\alpha-1}(1-p)^{\beta-1}}{B(\alpha, \beta)} dp \\ &= \frac{\binom{n}{k}}{B(\alpha, \beta)} \int_0^1 p^{k+\alpha-1} (1-p)^{n+\beta-k-1} dp = \binom{n}{k} \frac{B(k+\alpha, n+\beta-k)}{B(\alpha, \beta)} \end{aligned}$$

For convenience, it is common to reparametrize it as:

$$\begin{aligned} \mu &= \frac{\alpha}{\alpha + \beta} \\ \rho &= \frac{1}{\alpha + \beta + 1} \end{aligned}$$

so that the expectation and variance of the beta-binomial distribution are:

$$E(k|n, \mu, \rho) = n\mu$$

$$Var(k|n, \mu, \rho) = n\mu(1-\mu)[1 + (n-1)\rho]$$

In this form, μ corresponds to the estimate of p , and ρ corresponds to the extent of over-dispersion. Both μ and ρ values are between 0 and 1.

When we use beta-binomial distribution to model the RNA editing events in RNA-seq, n corresponds to the total number of reads overlapping with an RNA edit site and k to the number of reads with A-to-G mutations. In this scenario, the beta-binomial distribution is a better model for read counts than the binomial distribution since it takes the variability in mutation frequencies between biological samples into account. Under the null hypothesis, all samples have equal RNA editing level, and the edit frequencies are drawn from the same beta distribution $\pi(\mu_0, \rho)$. Under the alternative hypothesis, the samples expressing the MSI2-ADA fusion protein have a different RNA editing frequency than the control samples, and the frequencies come from two different beta distributions $\pi(\mu_1, \rho)$ and $\pi(\mu_2, \rho)$. Using the read counts at each RNA edit site from biological replicates, we maximized the likelihood for both the null and alternative hypotheses and then computed the p-value using a likelihood ratio test. The p-values from all sites were adjusted to control for false discovery rate (FDR) using a Benjamini-Hochberg correction. The statistical computation was performed using R packages *VGAM* and *bbmle*.

Significant sites were determined by filtering for FDR, using FDR < 0.05 for MOLM-13, FDR < 0.005 for LSCs and LSKs and FDR < 0.1 for HSPCs.

A target gene is retained if it has an expression level of at least 5 fpkm (in MSI2-ADA, MSI2-DCD and MIG) and at least one edit site with a significant differential edit frequency of at least 0.1 (differential edit frequency is edit frequency in MSI2-ADA subtracted to edit frequency by MSI2-DCD and empty vector MIG controls).

We have also added to the manuscript (first section of Results) a brief explanation as follows: “Importantly, to take into account the background editing by these controls, when calculating the actual edit frequency at each site, we reported the difference between the editing frequency of MSI2-ADA and that of MSI2-DCD and MIG (called differential edit frequency or diff.frequency).”

How do the authors define significantly different editing between cell types? Frequency of edit sites fluctuates more easily for sites that are not detected by many reads than those that have tens or hundreds of reads. Does a change from edit frequency of 0.4 to 0.6 mean a relevant change in binding? A number of representative examples of individual genes or binding sites showing a pile-up of edited and total (non-edited) reads in controls and test cell lines would be useful. Similarly for edit sites with significant differences between cell lines.

We thank the reviewer for this important point as we have strengthened the rigor behind this analysis and have now used a strict statistical test (beta-binomial test) to determine differences in edit frequencies in HSCs versus other populations and in the LSK and LSC analysis. These new results have now been added to the revised manuscript. We have replaced our original heatmaps with new heatmaps containing only significantly different sites between cell types (Figure 2E, Figure 3F and Supplemental Figure 3I). The new analysis strengthens our claims that a subset of targets is statistically more edited in HSCs versus other populations, and that LSCs have higher MSI2 binding than LSKs.

Do the authors cluster nearby edit sites in any way? ADAR linked to MSI2 via a flexible linker will presumably edit any suitable adenosine near the binding site, which may inflate the final number of binding sites. Is there a need to group edit sites in near proximity to formulate binding regions? Please discuss, and re-analyse if appropriate.

We thank the reviewer for this suggestion and have included this question into the revised manuscript. To decide on a suitable window size for clustering edit sites, we compared the enrichment of MSI2 motifs in windows of fixed size around significantly edited sites (“true sites”) compared to windows of the same size around non-significantly edited sites (“background”). Using Fisher’s test, we determined that +/-17bp is the largest window such that the motif enrichment was significantly greater around true sites compared to background. We therefore clustered nearby edit sites falling within this window size and found that the majority of clusters (87%) contain only single sites, suggesting that MSI2 mainly binds RNA at more dispersed sites (Supplemental Figure 1J and K). These data suggest that MSI2 binds at discrete

sites and ADAR doesn't edit multiple As near each binding event and the editing sites are therefore not inflated. Thus, we believe that these results justify our analysis approach.

(4) Control for endogenous RNA editing and ADAR expression:

With a method that relies on detection of editing on RNA, it is critical to be able to account for background that comes from endogenous RNA editing activities and sequencing errors. The authors use a catalytically dead version of MSI-ADA fusion as an appropriate control. I would like the authors to present more data on how these controls were analysed and interpreted, as this is not clear from the current manuscript. I would also like to see an experiment to show that endogenous ADAR activity is not different between the various cell lines the authors compare to rule out its impact on the observed differences in A-to-I editing.

As the reviewer rightly pointed out, the background editing by endogenous enzymes could be different between various cell types. Therefore, to take into account this background we used MSI2-DCD (the catalytically dead version of MSI-ADA) and empty vector (MIG) controls. To calculate edit frequency for each site, we subtracted the edit frequency in MSI2-ADA to that in MSI2-DCD and MIG controls. This was carried out for all cell types used in the study. To make it clearer in the revised manuscript, we have changed the term "edit frequency" to "differential frequency" (abbreviated *diff.freq* in the figures), which means the difference in frequency (the result of the subtraction) in MSI2-ADA and controls.

(5) Impact of over-expression of the MSI-ADA construct:

Do the authors think that over-expression of MSI-ADA construct might lead to MSI binding to sites it normally would not interact with? Is there a way to try to distinguish 'true' binding events from such artificial sites, for example by setting stringent criteria on replication and a high(er) threshold on edit frequency and total number of reads to support any give edit site? How sensitive is the method to possible differences in expression of the MSI-ADA construct when comparing different cell types? Please discuss potential pitfalls of MSI-ADA overexpression and any solutions.

We cannot completely rule out the possibility that the MSI2-ADA fusion might lead to binding to non-specific sites. This concern is not unique to the HYPERTRIBE approach as CLIP experiments could also have additional sites. Nevertheless, we performed multiple validations to show that editing events by MSI2 HyperTRIBE reflect true binding events. First, we looked to see if the editing sites are associated with any consensus sequence and found that they are enriched for known MSI2 motif (Figure 1E). Additionally, with any given editing event we can find a nearby MSI2 motif within 250bp (Figure 1F, cyan shape). This indicates that MSI2-ADA and MSI2 share recognition sequences. Second, to distinguish the "true" binding events and artificial binding sites by MSI2-ADA fusion we compared editing sites by MSI2 hyperTRIBE and MSI2 iCLIP peaks that were previously published. Since in MSI2 iCLIP (Rentas et al., 2016) MSI2-RNA interactions were captured using MSI2 antibody and no transgene overexpression was involved, they should reflect endogenous binding sites albeit with the potential for false positive sites from the IP. As shown in Figure 1F, the editing occurred either on or near sites that

were directly bound by MSI2 identified by iCLIP, suggesting that vast majority of MSI2-ADA edit sites are reproducible and independently verified binding sites for MSI2.

Third, we hypothesized that if the targets are specific for MSI2, they would be affected by MSI2 depletion. Therefore, we sought to see the correlation between the binding target genes and genes with differential expression upon MSI2 depletion in leukemia cell lines using GSEA analysis. Results show that top genes with highest editing frequency positively correlated with genes up-regulated upon MSI2 depletion, suggesting that the targets determined by MSI2-ADA are also directly regulated by MSI2. Fourth, in our revised experiments with a mutant RRM or deleted RRM the vast majority of the MSI2 binding sites are lost, demonstrating that MSI2-ADA editing is dependent on MSI2 RNA binding, hence reflects “true” MSI2 RRM dependent binding sites.

Importantly, we set stringent criteria for replication, edit frequency and number of reads. Details are in the Methods section. Briefly, significant sites are determined if it has a coverage of at least 5 FPKM (in MSI2-ADA, MSI2-DCD and MIG) and has differential frequency of at least 0.1 (diff.freq of a site is the edit frequency in MSI2-ADA minus the average edit frequency in MSI2-DCD and MIG and has to be greater than or equal to 0.1). Additionally, we used a statistical test to determine the FDR-corrected p-value for each site, which considers variation across biological replicates.

(6) Categorisation of target (edited) RNAs by cell type:

I find the visualization and grouping based on editing frequencies in Figures 2E and 3F misleading. Is there a biological or technical basis for highlighting a specific range in editing over others? For example, is there a biological or technical reason to say that a change from edit frequency 0.4 to 0.6 is much more indicative of a drastic change in binding than a change from 0.2 to 0.4 or from 0.6 to 0.8? This is what Figure 3F conveys visually due to the colour scale chosen to display edit frequencies. This visual threshold is also set differently in the two figures: In Figure 2F the chosen colour scheme emphasises differences in 0-0.2 range over 0.2-0.8, and in Figure 3F it emphasises differences in 0.4-0.6 range over other similarly sized ranges. Based on the evidence provided, I am not convinced that the grouping of genes into the presented clusters is meaningful. Please display this data on a continuous colour scale that gives a truthful visual impression of differences in edit frequencies, and discuss if there is a way to assess what level of changes in editing are indicative of true differences in binding.

We have now applied a beta binomial statistical test to evaluate the significance of the differential edit frequency for each site across compared cell types. We report results using diff.frequency relative to cell-type specific controls. The different color scales chosen in Figure 2E and 3F are due to the fact that the diff.frequency in HSPCs is generally lower than that in LSK and LSC.

(7) Functional impact of MSI2 binding:

The manuscript does not go very far in explaining the functional impact on MSI2 binding at a molecular level and on cell-type differences. It is interesting that MSI2 binding does not seem to have a connection with RNA expression levels (Figs 2E and 3F), but this begs the question – what impact does MSI2 binding have? The authors confirm that MSI2 depletion impact protein

levels of two previously known target genes and one novel one, suggesting a role for MSI2 in translation. I would hope to see a more proteome-wide analysis of the impact of MSI2 binding to strengthen this point in a journal like Nature Communications. I would also like to see an experiment that directly shows that MSI2 binding affects translation (e.g. by mutating the binding site and observing target RNA & protein abundance).

The main goal of this study is to demonstrate that our optimized HyperTRIBE method is applicable for RNA target identification in rare cells, which we think is of great interest to the broader scientific community in the stem cell field. Secondly, using this approach we determined elevated binding activity of the RBP MSI2 in leukemia stem cells compared to normal blood stem cells and this results in differential translational regulation of known and newly identified targets. We have identified and validated in this study the translational regulation of MSI2 on 5 targets of which 2 are newly identified. We acknowledge the reviewer's comment that the latter point, about proteome-wide analysis, could be expanded to a study on general mechanism of MSI2 regulation. However, while we think this is definitely a highly desired experiment, it is out of the scope of our manuscript. Moreover, it is technically impractical to perform proteomics in rare stem cell populations and only likely to capture the most abundant proteins making it difficult to draw conclusions about MSI2 function in these cell types. However, to confirm that MSI2-HyperTRIBE sites are indeed functional regulation sites we have performed luciferase reporter assay using the original and mutated 3'UTRs for *Hoxa9* and *Myb*. The results and figures are in the revised manuscript. Upon *Msi2* depletion or overexpression the reporter signal was reduced or increased, respectively. Importantly these effects were abolished when the sites were mutated. Furthermore, we have shown that *Msi2* depletion in LT-HSCs results in reduced BRCC3 abundance and in LSCs led to a reduction of HOXA9, IKZF2 and MYC without altering their mRNA level in the endogenous setting. Together these data suggest that MSI2 regulates translation of these mRNA targets through their identified binding sites.

(8) Impact of artificial editing on the transcriptome:

A-to-I editing exist in cells to regulate RNA life cycle. Is there a risk that high expression of exogenous ADAR, used here to identify binding regions, artificially influences the expression and processing of those RNAs? Please discuss in text.

We have addressed this issue in the revised manuscript by providing differential expression (DESeq2) analysis for cells expressing MSI2-ADA compared to those with empty vector (MIG) (Supplemental Figure 1G, Supplemental Figure 3L and O). Our analysis shows that there is little change in the transcriptome of MOLM13, LSK and LSC expressing MSI2-ADA after 48 hours of transduction. For *in vivo* HyperTRIBE in HSPCs, which took 7 weeks for transplantation and engraftment of cells expressing MSI2-ADA, we observed dramatic changes in transcriptome of LT-HSC and ST-HSC but not MPP2 and MPP4. The analysis is attached here to the response. Of genes significantly changed upon MSI2-ADA expression, the majority are due to MSI2 overexpression and only 27 and 36% are due to ADAR editing in LT-HSCs and ST-HSCs, respectively.

Differential expression analysis of MSI2 overexpression in 4 HSPCs populations. Red dots represent genes that have expression significantly different in MSI2-ADA compared to MIG control (adjusted $p < 0.05$). X-axis is \log_2 fold change of gene expression in cells expressing MSI2-ADA vs. cells expressing empty vector MIG.

INTERMEDIATE POINTS:

(1) Previous research on RNA-binding proteins in context of stem cell:

Page 1 lines 16-18: “However, the aspect of cell-context dependent activity of RBPs during normal stem cell differentiation and transformation to malignancies has never been addressed” and page 2 lines 43-43: “However, whether RBPs may have cell-type specific activity between different cellular states of normal stem cell differentiation or between normal and transformed cells has never been addressed”. A good number of studies have looked at RNA-binding proteins in stem cells, differentiated cells, and malignant cells, although many of these studies may not have taken a high-throughput approach. I agree with the authors with the importance of studying the activity of RBPs in a cell-type specific manner during cell differentiation, but I would recommend they tone down this statement and discuss and cite the work done in this realm. This could include, for example, some the following original research and review articles:

- Li and Izpisua Belmonte. Deconstructing the pluripotency gene regulatory network. *Nat Cell Biol.* 2018. PMID: 29593328
- Yang et al. Imp and Syp RNA-binding proteins govern decommissioning of Drosophila neural stem cells. *Development.* 2017 PMID: 28851709
- Hayakawa-Yano et al. An RNA-binding protein, Qki5, regulates embryonic neural stem cells through pre-mRNA processing in cell adhesion signaling. *Genes Dev.* 2017. PMID: 29021239
- Ju Lee et al. A post-transcriptional program coordinated by CSDE1 prevents intrinsic neural differentiation of human embryonic stem cells. *Nat Commun.* 2017. PMID: 29129916
- Degrauwe et al. IMPs: an RNA-binding protein family that provides a link between stem cell maintenance in normal development and cancer. *Genes Dev.* 2016. PMID: 27940961
- Wurth and Gebauer. RNA-binding proteins, multifaceted translational regulators in cancer. *Biochim Biophys Acta.* 2015. PMID: 25316157

• Kwon et al. The RNA-binding protein repertoire of embryonic stem cells. Nat Struct Mol Biol. 2013. PMID: 23912277

We thank the reviewer for the comment on this point. Most of these studies, however, looked at embryonic stem cells, pluripotent stem cells, neural stem cells that isolated from embryos that exists in large numbers. Although study by Palanichamy et al. (2016) looked at hematopoietic stem cells they identified RBP targets using leukemia cell lines. In addition, the study by Degrauwe et al. (2016) performed PAR-CLIP using 10^8 glioblastoma stem cells. Importantly, as the reviewer rightly pointed out that even though some of these studies investigate RBP functions in normal and malignant stem cells (Degrauwe et al. 2016) they were not done systematically using high throughput approaches.

We have added these references to the introduction and discussion and want to focus on the concept that this approach investigates RBP targets in rare cells such as adult stem cells.

(2) ‘Gene and Drug’ enrichment analyses:

Figures 2F, 3H-K: Please elaborate on how the listed significant Gene&Drug signatures indicate stemness or differentiation, and what the stemness/differentiation gradient in Figure 2F is based on. The terms seem to related to specific knock-downs and knock-outs in different cell types, for example to previously done MSI2 experiments. This is perhaps technically encouraging, but I am wondering if this analysis tells more about shared RNA targets than stemness/differentiation per se. Admittedly, I am not familiar with the Gene&Drug signatures resource, so perhaps an expanded description of the resource and method would go a long way here. Would a Gene Ontology (GO) enrichment analysis give a more straight-forward indication of whether the targets are involved in stemness and differentiation? Could the authors try this?

We originally overlaid the gradient of stemness and differentiation based on the combine scores in ENRICH analysis. HyperTRIBE targets in LT and ST-HSC are enriched for the genes significantly changed in various indicated knockout or knock-downs in HSCs and LSCs, suggesting that these targets are important for maintaining the functions of these stem cells. Details on the description of this pathway enrichment tool are in the reference which we cited in the main text and Methods. We have removed the gradient in Figure 2F as it was purely descriptive. We have now provided Gene Ontology (Molecular Function and Biological Processes) analysis in the Supplemental Figures 2I, J, 3P and supplemental table S4.

(3) Technical details in text:

Please include more technical details in the main manuscript. For example, please state clearly in the text how many replicates were analysed for each experiment. Minimum of 2 is required for RNA-seq based experiments, ideally 3 (or more).

We have expanded our technical details and have added missing data in the figure legends as well as in the methods. All experiments were carried out with at least 3 replicates, except for the *in vivo* HyperTRIBE in HSPCs which was done with 2 replicates.

(4) Details on data availability:

Please indicate clearly the database and IDs where the raw and processed data will be publicly available upon publication (for example, Gene Expression Omnibus or ArrayExpress).

We have deposited our code onto Github at [git@github.com:DiuTTNguyen/MSI2_HyperTRIBE_codes.git](https://github.com/DiuTTNguyen/MSI2_HyperTRIBE_codes.git) and our raw and processed data onto GEO with the accession number GSE132949. We will make this information publicly available upon publication.

MINOR POINTS:

(1) Figure 1 panel D:

Does this graph take into account the relative length of these features? 3'UTRs are probably considerably longer than CDS and 5'UTR regions for a large fraction of genes. To show preference for 3'UTRs, one would ideally take their relative length into account.

This is an interesting point. We have not taken into account the relative length of these features. However, as discussed in the text previous studies have reported the preference of MSI2 binding to 3'UTR regions. Another issue is that there are DNA compositional differences between 5'UTR, CDS, and 3'UTR regions, with 3'UTRs being the most A-rich. We confirmed that the editing frequency at As in the 3'UTR is an order of magnitude higher than in the CDS (z score=477.2592) and significantly higher in 3'UTRs compared to all regions ($p < 2.2e-16$, Fisher's exact test). This analysis suggests that MSI2 prefers to bind to 3'UTR region regardless of the number of As.

(2) Figure 1 panel E:

Why is the MSI1 included in the label?

This is because MSI1 and MSI2 are known to share the same motif. We have removed MSI1 to avoid the confusion.

(3) Figure 1 panel F:

I am unclear on whether the distance is calculated from the motif (motif-to-hyperTRIBE site and motif-to-iCLIP site) or the HyperTRIBE binding site (hyperTRIBE-to-motif and hyperTRIBE-to-iCLIP). Please clarify. Either way, please make clear that that this plot does not indicate that iCLIP identifies the binding sites less precisely than HyperTRIBE.

The cyan peak is the distance from the edit site to the nearest MSI2 motif (found in HOMER *de novo* motif analysis) and the yellow peak is the distance from the edit site to the iCLIP peak. We have clarified in the figure legend. We did not claim this plot indicated iCLIP is less precise than HyperTRIBE. In fact, as discussed in the text we used iCLIP data as a benchmark to assess the accuracy of RNA target identification by HyperTRIBE.

(4) Figure 1 panel G:

Could the authors please clarify what is plotted here and how to interpret the graph? Are the 255 genes on the x-axis? The y-axis seems to be cut off at the bottom. What does 'NES' stand for? Does the FDR cutoff refer to the hyperTRIBE done in this manuscript, or the dataset it is being compared to (Kharas et al. 2010)? What does the bottom part of the graph represent (black barcode), and what does the red and blue scale mean? I am also confused about the placement of the labels 'Ctrl', 'MSI2 knock-down UP' and 'MSI2 knock-down'

This is a Gene Set Enrichment Analysis (GSEA) plot. Details can be found here https://software.broadinstitute.org/cancer/software/gsea/wiki/index.php/Main_Page Essentially we ranked the top genes with highest diff.freq in HyperTRIBE. We then compared this list to the list of genes with ranked log2fc (MSI2 Ctrl/KO) (RNAseq data). The enrichment score (on the y-axis) is added if a gene from the HyperTRIBE list is found in the RNA-seq list and is subtracted if not. NES stands for normalized enrichment score (please see the link for detailed explanation). FDR is referred to the exact FDR calculated by the GSEA software (please see the link for further details how it is computed).

(5) Figure 3 panel J legend:

The description "Gene expression (GE) independent RNA-seq Gene and Drug signatures" is a little confusing to me. Could it be clarified and simplified to "Significantly enriched Gene and Drug signatures based on ENRICH analysis of shared MSI2 targets in LSKs and LSCs"?

We have clarified in the gene expression independent targets in the legend of Figure S3E in the revised manuscript. Figure 3J shows the ENRICH analysis of a subset of shared (LSK and LSC) MSI2 targets which expression (or transcript abundance) in LSC is lower or equal to LSK. This subset is majority of the shared targets as shown in Figure 3E (pink bar).

(6) Figure 3 panel K legend:

Similarly to legend J, I find the description "GE independent signature of RNA-seq" unclear. Please modify.

Similar to point 5, this graph shows signatures of LSK and LSC unique targets which are not biased in their expression towards LSC (specified by log2FC and p-values in DESeq2 in the text). Ultimately, we wanted to point out that the enriched pathways that we observed is not due to higher RNA expression of these targets in LSC.

(7) Figure 4 panels A-C:

Could the authors show these binding sites with respect to their real position along the (exonic regions) 3'UTR rather than in an abstracted order? Something like a nice version of a genome-browser screenshot. It would be informative for the reader to see if and how the binding sites cluster along the 3'UTR. It would also be very informative to see in this plot, perhaps on a

separate track, the edit frequency in control cells, as well as to total number of reads the frequency calculation is based on. Does the significance test and 'ns' label refer to comparison between LSK and LSC at a given site? Please clarify in the legend.

We have updated these figures in the revised manuscript to include real positions along the 3'UTR region. We have also added the predicted MSI2 motif to the graphs. The edit frequency (now more accurately described as differential frequency) for each site is already normalized to the edit frequency in the control cells. We have added the beta binomial to test the significance of each site if shared between LSK and LSC and have provided the sites that were significantly more edited in one cell type versus the other. Importantly, this method looks at the gain of variant of a nucleotide on RNAs (editing events), and not enrichment of recovered RNA reads by immunoprecipitation (as in other CLIP methods). Therefore, it does not produce the typical peak track seen with other methods. Snapshots of bam files are attached.

As seen here, the pile up reads and coverage tracks (snap shot from IGV) do not show all editing events (mutations). These will be called from the VariantCaller, which are several steps after producing bam files.

(8) Sharing the code so that the scientific community can build upon it:

I would like to encourage the authors to make all the code written to analyse these data available, for example on GitHub or as a supplementary to the manuscript. As this may not be Nature Communications policy, I do not consider it necessary for publishing this manuscript. The authors mention the R script for distance-to-motif analysis is available on GitHub. Please provide the URL.

Our code for this manuscript can be found at

GitHub link: [git@github.com:DiuTTNguyen/MSI2_HyperTRIBE_codes.git](https://github.com/DiuTTNguyen/MSI2_HyperTRIBE_codes.git)

Our data have been deposited on GEO with the accession number [GSE132949](https://www.ncbi.nlm.nih.gov/geo/query/acc.cgi?acc=GSE132949).

Reviewer #3 (Remarks to the Author):

The manuscript by Nguyen et al employs the HyperTRIBE approach to reveal MSI2 RNA binding network in haematopoietic stem cells (HSCs) and leukaemic stem cells (LSCs). Multiple RNA binding proteins have been found to play key regulatory roles in normal haematopoiesis and leukaemic transformation. However, small cell numbers have always been a limiting factor in identifying their target transcripts at high resolution. In this manuscript, the authors have

optimised and used HyperTRIBE method, which so far has worked in Drosophila cells but has never been employed in mammalian cells. This paper not only provides a substantial technological advance but also reveals important insights into how MSI2 functions in normal and malignant haematopoiesis.

My comments and suggestions are listed below.

- 1) Given the novelty of this method in mammalian cells, it would be important that the authors include the full sequences of MSI2-DCD and MSI2-ADA for the benefit of the research community.

We have provided the full sequences of these fusions in the revised manuscript. We will deposit the vector into Addgene for the scientific community since we have been asked by many researchers.

- 2) It is noted that human MSI2 was used. What was the rationale behind using human MSI2 if the goal was to perform the study in murine cells? Are mouse and human MSI2 proteins conserved?

Yes, mouse and human MSI2 proteins are highly conserved. Pairwise sequence alignment by EMBOSS Water (developed by EMBL-EBI) shows they are 94.5% similar.

- 3) Human AML cells were used to optimise the HyperTRIBE approach. This gives the opportunity to compare MSI2- interacting transcripts between human and mouse leukaemic cells at high resolution. What was the degree of an overlap between MSI2-binding transcripts between human and mouse cells? Are the targets conserved? What are they?

Overlapping MSI2 targets in the human cell line MOLM-13 (2056 genes) and the mouse LSCs (4163 genes) shows that over 60% of the human cells (1248 in total 2056) are shared with the mouse cells. The overlapping is attached for the reviewer's reference. However, note that MOLM-13 is a cell line whereas LSCs are leukemic stem cells (c-kit high leukemia cells).

Overlapping HyperTRIBE target genes in human cells MOLM-13 and in murine LSCs

- 4) Regarding data presented in Figure 2 – these are very exciting results highlighting the feasibility of high-resolution editing in small numbers of cells. Do the authors have data

indicating that HSCs were sorted from recipients with multilineage reconstitution?

We did not perform characterization of all lineages in the recipient mice at the time of sorting cells for RNA-seq as we were interested only in the engraftment at the level of the HSC and MPPs.

5) The authors conclude “Msi2 depletion was likely altering SMAD3 translation specifically in LT-HSCs compared to ST-HSC and MPPs (Park et al., 2014).” They go on to state: “These data indicate that despite similar abundance of MSI2 and its targets, MSI2 can differentially control its targets’ expression during hematopoietic differentiation...”. Does MSI2 control translation of other direct HSCs-specific targets identified by the HyperTRIBE approach?

We now have used a stringent statistical test (see the Methods description of the beta binomial test) to test the differential editing of MSI2-ADAR, which reflects differential binding of MSI2, at targets unique for LT-HSC compared to ST-HSC and MPPs as well as other groups of genes that are unique to other populations. We have provided the data in Figure 2E of the revised manuscript. Among these, *Brcc3* has been reported to mutate in AML and MDS, and these mutations are associated with clonal hematopoiesis suggesting its function in HSCs. Therefore, we selected this target to further validate by immunofluorescence. We found that BRCC3 protein level is significantly reduced in LT-HSC but not in ST-HSC and MPPs upon Msi2 deletion. Our previous data shows no change in mRNA level, suggesting that *Brcc3* is a novel translational target of MSI2 in HSC.

Furthermore, although *Smad3* was previously identified, this study provides a new insight in that *Smad3* binding and regulation by MSI2 is differential in HSCs versus MPPs, which was not possible using other techniques.

6) In the discussion, that authors state: “Importantly, our results demonstrate that RBP-RNA interactions are highly cell- context dependent. We revealed that MSI2 has differential binding activity at different states of HSPCs and in LSCs in a target gene expression independent manner. It would be very informative to expand this section and offer some potential explanation for the cell-context dependency of MSI2 and the stronger editing in LSCs vs HSCs. Furthermore, why does MSI2 controls HOXA9, IKZF2 and MYC protein expression in LSCs but not HSCs? It would be desirable to discuss these data in more detail.

We thank the reviewer for these interesting questions. We have provided further discussion in the revised manuscript as follows. “Furthermore, it remains to be elucidated (1) how MSI2 achieves more binding to mRNA targets in LSCs even without up-regulating MSI2 expression; and (2) why MSI2 controls protein expression of its targets (e.g *Hoxa9*, *Ikzf2* and *Myb*) in LSCs but not in normal HSPCs. One possibility is that other RBPs that share a similar binding motif might compete for the same binding sites with MSI2 in LSKs. Alternatively, post-translational modifications on MSI2 or other RBPs could result in the increased binding. Moreover, multiple RBP-driven regulation pathways, including MSI2’s may coordinate to control translation process of their shared targets. Cancer cells often alter or lose multiple pathways and thus might be dependent on the MSI2 regulation. Therefore, LSCs recruit more MSI2 to its targets rather than different RBPs as in normal LSKs. As a consequence, the regulation of the target expression is

now more dependent on MSI2. Regardless of the exact mechanism, our data support a leukemia specific role for MSI2 and provide further rationale for targeting MSI2 in leukemia cells in patients that have equivalent expression of MSI2 as compared to normal cells.”

Minor points:

1) Figure 2A – schematic - should be “irradiation” not “irridiation”.

We have made the appropriate change in the revised manuscript.

1) For consistency, all genes and transcripts should be italicised in the figures.

We have made the appropriate change in the revised manuscript.

REVIEWERS' COMMENTS:

Reviewer #3 (Remarks to the Author):

The authors have done a great job revising the manuscript.
I think it now meets the high standards of Nature Communications.

Reviewer #4 (Remarks to the Author):

Nguyen et al. have done some major improvements to the manuscript since the previous round of review including a better statistical analysis (and description) of the binding sites, improved technical description of how the analyses were done and code and data availability, three additional HyperTRIBE experiments as controls, and some more functional analyses. Overall, I am satisfied with the changes and the effort the authors have put in, and in my opinion this article is suitable for publication in Nature Communications once the authors have addressed the below minor points.

(1) I find the authors answers to my questions on cell numbers and sensitivity (MAJOR 1), biological novelty (MAJOR 2), and differential expression analysis on the impact of exogenous ADAR expression (MAJOR 8) informative and I would like the authors to provide this insight (and claims) in the main text and/or supplementary as appropriate.

(2) I would like the authors to address the issues with Figure 1G (e.g. what the x and y axis are, the acronyms, etc please see the details in my original review). The authors could also include a short version of the description provided in the rebuttal in the figure legend or elsewhere in the manuscript. I am ok with the contents of the analysis but the representation needs to be improved to help the reader understand what this graph shows.

Reviewer #5 (Remarks to the Author):

The authors have adequately addressed my comments and I have no further criticisms to make.

Response to reviewers' comments

Reviewer #3 (Remarks to the Author):

The authors have done a great job revising the manuscript. I think it now meets the high standards of Nature Communications.
Thank you.

Reviewer #4 (Remarks to the Author):

Nguyen et al. have done some major improvements to the manuscript since the previous round of review including a better statistical analysis (and description) of the binding sites, improved technical description of how the analyses were done and code and data availability, three additional HyperTRIBE experiments as controls, and some more functional analyses. Overall, I am satisfied with the changes and the effort the authors have put in, and in my opinion this article is suitable for publication in Nature Communications once the authors have addressed the below minor points.

(1) I find the authors answers to my questions on cell numbers and sensitivity (MAJOR 1), biological novelty (MAJOR 2), and differential expression analysis on the impact of exogenous ADAR expression (MAJOR 8) informative and I would like the authors to provide this insight (and claims) in the main text and/or supplementary as appropriate.

We have edited the revised manuscript to include all these points in the discussion.

(2) I would like the authors to address the issues with Figure 1G (e.g. what the x and y axis are, the acronyms, etc please see the details in my original review). The authors could also include a short version of the description provided in the rebuttal in the figure legend or elsewhere in the manuscript. I am ok with the contents of the analysis but the representation needs to be improved to help the reader understand what this graph shows.

We have edited the figure legend for clearer description.

Reviewer #5 (Remarks to the Author):

The authors have adequately addressed my comments and I have no further criticisms to make.

Thank you.